# GIC: Gaussian-Informed Continuum for Physical Property Identification and Simulation

**Junhao Cai**[1*]    **Yuji Yang**[2*]    **Weihao Yuan**[3†]    **Yisheng He**[3]
**Zilong Dong**[3]    **Liefeng Bo**[3]    **Hui Cheng**[2]    **Qifeng Chen**[1]
[1]The Hong Kong University of Science and Technology, [2]Sun Yat-sen University, [3]Alibaba Group
[*] Equal contribution, order determined by coin toss. [†] Corresponding author.

## Abstract

This paper studies the problem of estimating physical properties (system identification) through visual observations. To facilitate geometry-aware guidance in physical property estimation, we introduce a novel hybrid framework that leverages 3D Gaussian representation to not only capture explicit shapes but also enable the simulated continuum to render object masks as 2D shape surrogates during training. We propose a new dynamic 3D Gaussian framework based on motion factorization to recover the object as 3D Gaussian point sets across different time states. Furthermore, we develop a coarse-to-fine filling strategy to generate the density fields of the object from the Gaussian reconstruction, allowing for the extraction of object continuums along with their surfaces and the integration of Gaussian attributes into these continuums. In addition to the extracted object surfaces, the Gaussian-informed continuum also enables the rendering of object masks during simulations, serving as 2D-shape guidance for physical property estimation. Extensive experimental evaluations demonstrate that our pipeline achieves state-of-the-art performance across multiple benchmarks and metrics. Additionally, we illustrate the effectiveness of the proposed method through real-world demonstrations, showcasing its practical utility. Our project page is at https://jukgei.github.io/project/gic.

## 1 Introduction

Identifying the physical properties of objects (i.e., system identification) is essential for numerous applications such as games, digital twins, and robotic manipulation [1–3]. Although humans can intuitively deduce the underlying physical properties with a single glance when the object undergoes deformation, estimating the properties with only visual observations remains challenging for computational perceptual algorithms.

To tackle this challenge, many established methods [4–6] adopt the assumption of elastic material [7] and perform physics-based modeling based on mass-spring systems (MSS) or finite element method (FEM) to model and simulate the dynamics of the objects. Such an assumption inevitably restricts the ability to simulate more general types beyond elastic materials, such as fluids or granular media. Another problem of previous methods lies in that many methods [8–10] require the ground-truth full knowledge of object geometry for the identification, which limits their practicality. Some subsequent methods [5, 4] turn to recover the geometries and physical properties from observations in a decoupled manner. Specifically, these methods first extract object geometries by making use of stereo observations or dynamic neural reconstruction [11] from RGB video sequences, and then perform simulation directly on the point clouds or after the tetrahedral mesh conversion. While these methods introduce explicit geometries to guide the estimation of physical properties, the noisy reconstruction results usually lead to degraded system identification performance.

38th Conference on Neural Information Processing Systems (NeurIPS 2024).

Recently, PAC-NeRF [12] integrates neural radiance fields (NeRF) [13] with a continuum dynamic model to tackle the above problems. The object geometries and physical properties are captured in a unified framework. Despite its effectiveness, this method possesses two limitations. Firstly, the implicit shapes represented by NeRF often lead to inferior geometries, which might cause inaccurate trajectories during simulation. Secondly, PAC-NeRF renders the novel views of deformed objects based on the appearance radiance field reconstructed from the static scene, which might introduce texture distortion, particularly when objects undergo significant deformations, resulting in discrepancies between the rendered and the observed images [14].

To address these limitations, this paper proposes a novel hybrid solution based on 3D Gaussians [15, 16] and material point method (MPM) [17, 18]. The core strength of this work is that we make use of both 3D shapes from dynamic 3D Gaussian reconstruction and 2D shapes rendered by the Gaussian-informed continuum for physical property estimation.

To generate more precise shapes to reason physical property, we first propose a *motion-factorized dynamic 3D Gaussian network* to conduct dynamic scene reconstruction. We then extract the continuum from the recovered 3D Gaussians at each frame by leveraging a *coarse-to-fine filling strategy* to generate the density field of the object progressively. The resulting density fields can be used to sample continuum particles for simulation and extract object surfaces as 3D-shape supervision in physical property estimation. To eliminate the appearance distortion caused by large deformation in PAC-NeRF, we further assign Gaussian attributes to the continuum particles where the opacity and scale attributes are evaluated from the density field. Such *Gaussian-informed continuum* are able to render object masks during simulation, which can be regarded as a 2D-shape representation to guide the estimation and effectively avoid using inferior rendering results for learning physical properties.

To demonstrate the superiority of the proposed method over other baselines, we conduct three types of experiments, including evaluations of physical properties, dynamic reconstruction, and future state simulation. We also demonstrate a real-world application in digital twins and robotic manipulation, showing the applicability of the proposed method in real-world scenarios.

Our contributions are summarized as follows.

- We propose a novel hybrid pipeline that takes advantage of the 3D Gaussian representation of the object to both acquire 3D shapes and empower the simulated continuum to render 2D shapes for physical property estimation.

- We propose a novel dynamic 3D Gaussian framework with motion factorization to achieve more precise dynamic reconstruction. We also propose a coarse-to-fine filling strategy to generate the density field of the object, which can be utilized to extract object surfaces and obtain Gaussian-informed continuum particles.

- Extensive experiments show that our pipeline attains state-of-the-art performance on existing benchmarks with a wide range of metrics. We also present a real-world demonstration to show the efficiency of the proposed method.

## 2 Related Work

**Dynamic reconstruction**. Reconstructing dynamic scenes from monocular or multi-view video(s) is a long-standing problem in the computer vision community [19, 20]. Previous works exploit neural implicit representation [21, 22] for non-rigid reconstruction. These methods either reconstruct the scene in a frame-wise manner [23, 24] or maintain a canonical shape and model the deformation with a neural network [25, 26, 11, 27]. While effective for novel view synthesis, these methods often require extensive training time and can result in noisy deformations owing to the implicit representation, which may compromise the utility of the recovered geometries for physical property estimation [12]. Recent progress in 3D Gaussian Splatting (3DGS) technique [15] stands out to be a prevalent method for 3D reconstruction and novel view synthesis because of the abilities of explicit shape modeling and extremely fast view rendering. Similar to non-rigid NeRF, many follow-up works extend the 3DGS into 4D by treating each frame separately [28] or decomposing a scene into a canonical 3D Gaussian point cloud and a deformation model that warps the canonical shape into a specific scene [16, 29, 30]. In this paper, we draw upon these prior studies [16, 29] and propose a novel motion-factorized dynamic 3D Gaussian network to achieve better performance on reconstruction and novel view synthesis.

**System identification**. Understanding the physics laws of the 3D world is beneficial for simulation [31–35, 6] and manipulation [2, 3, 36–38]. However, unveiling these properties from visual information is an extremely difficult task due to the ambiguity introduced by incomplete observation and the high degrees of freedom of the scene. Early works [39, 40] study the problem by learning physical properties via interactions. With recent improvements in differentiable physics simulation [17, 18, 41–45], many methods turn to evaluate the physical properties by comparing the rendering results with 2D ground truth given the prior knowledge about the object geometry. VEO [5] presents a differentiable simulator to learn patterns from 4D reconstruction and force-displacement measurements. Another approach [4] eliminates the dependence of captured forces by proposing an iteration framework between deformation tracking and parameter optimization. While these methods demonstrate promising results, the inferior reconstruction might lead to degraded performance, and the assumption of elastic material restricts the applicability. PAC-NeRF [12] instead proposes a single framework to recover both the unknown geometry and physical properties of deformable objects from multi-view video sequences. However, the inferior geometries and blurry rendered images might have detrimental effects on physical property reasoning. In this work, we adopt MPM as our simulation framework following the approach used in PAC-NeRF due to its ability to simulate a variety of materials [6, 46–48]. Unlike previous approaches, we utilize dynamic 3D Gaussians to reconstruct explicit 3D geometries and generate simulatable continuum particles. Furthermore, we enhance the particles with Gaussian attributes, facilitating the rendering of 2D shapes, and thereby improving physical parameter estimation.

## 3 Preliminary

In this section, we briefly review the core idea of 3D Gaussian Splatting (3DGS) [15] and introduce its point-based alpha blending to render depth maps and foreground masks. Typically, 3DGS utilizes 3D Gaussians, each defined by a central point $\mu_0$, a covariance matrix $\Sigma_0$, a density value $\sigma$, and a color attribute $c$, to efficiently render images from specific viewpoints. Each point is denoted as

$$G(x) = \exp(-\frac{1}{2}(x - \mu_0)^T \Sigma_0^{-1} (x - \mu_0)), \tag{1}$$

where $\Sigma_0$ can be factorized as $\Sigma_0 = R_0 S_0 S_0^T R_0^T$, in which $R_0$ is a rotation matrix represented by a quaternion vector $r_0 \in \mathbb{R}^4$, and $S_0$ is a a diagonal scaling matrix characterized by a 3D vector $s_0 \in \mathbb{R}^3$. If we consider isotropic Gaussian representation, the scaling matrix can be written as $s_0 I$, where $s_0$ is a scalar and $I$ is the identity matrix. When performing splatting, the 3D Gaussians are projected into 2D with the covariance matrix defined as $\Sigma_0' = JW\Sigma_0 W^T J^T$, where $J$ is the Jacobian of affine approximation of the projective transformation [49], and $W$ is the viewing transformation matrix. The rendered color $I(u)$ with its foreground mask $A(u)$ at pixel $u$ are then evaluated by integrating $N$ ordered slatted Gaussians via the point-based alpha blending. Since the depth of each Gaussian point at a specific view can be obtained according to its transformation matrix, we can further render the depth map $D$ using the same blending method [16, 50], as

$$I(u) = \sum_{i \in N} T_i \alpha_i c_i, \qquad A(u) = \sum_{i \in N} T_i \alpha_i, \qquad D(u) = \sum_{i \in N} T_i \alpha_i d_i, \tag{2}$$

where $T_i = \prod_{j=1}^{i-1}(1 - \alpha_j)$ is the accumulated transmittance, $\alpha_i$ is the probability of termination at point $i$, and $d_i$ is the depth of the Gaussian point at the specific view.

## 4 Method

### 4.1 Problem Definition and Overview

In this work, we aim to reconstruct the geometries and the physical properties of various object types from multi-view videos. Formally, given a set of video sequences $\{V_i | i = 1...n\}$ with moving object and the corresponding camera extrinsic and intrinsic parameters $\{(T_i, K_i) | i = 1...n\}$, the goal of this task is to recover the explicit geometries of the object represented by continuum particles $P(t)$ and its corresponding physical parameters $\Theta$ (e.g., Young's modulus $E$ and Poisson's ratio $\nu$ for elastic objects). We follow the assumption in PAC-NeRF and PhysGaussian [12, 51] that the object types (e.g., elastic, granular, Newtonian/non-Newtonian, plastic) are known and the physical phenomenon follows continuum mechanics [17, 52].

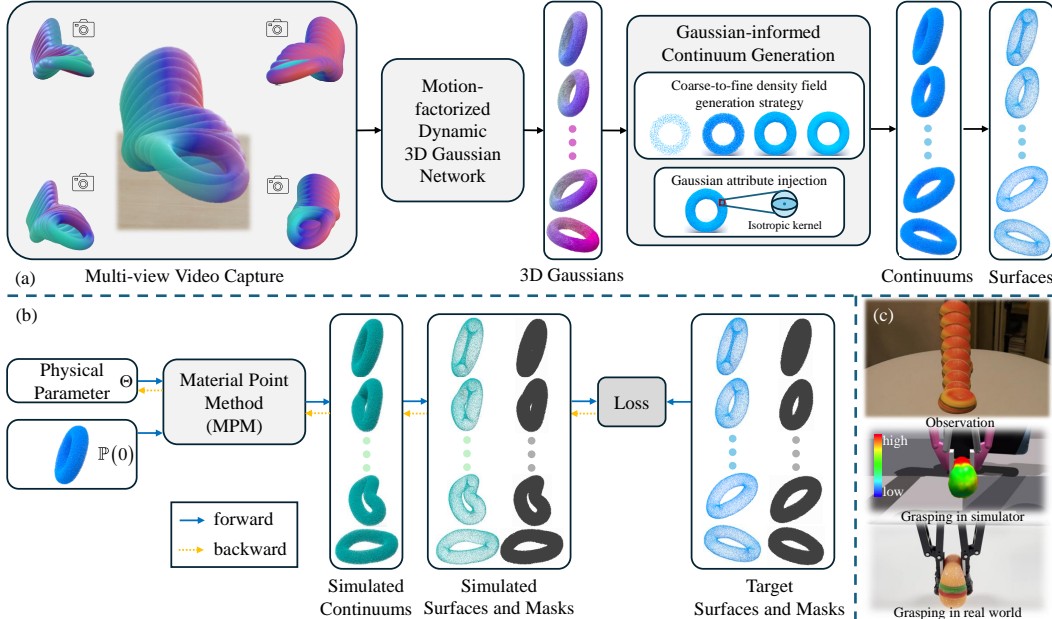

Figure 1: Overview. (a) **Continuum Generation:** Given a series of multi-view images capturing a moving object, the motion-factorized dynamic 3D Gaussian network is trained to reconstruct the dynamic object as 3D Gaussian point sets across different time states. From the reconstructed results, we employ a coarse-to-fine strategy to generate density fields to recover the continuums and extract object surfaces. The continuum is endowed with Gaussian attributes to allow mask rendering. (b) **Identification:** The MPM simulates the trajectory with the initial continuum $\mathbb{P}(0)$ and the physical parameters $\Theta$. The simulated object surfaces and the rendered masks are then compared against the previously extracted surfaces (colored in blue) and the corresponding masks from the dataset. The differences are quantified to guide the parameter estimation process. (c) **Simulation:** Digital twin demonstrations are displayed. Simulated objects (colored by stress increasing from blue to red), characterized by the properties estimated from observation, exhibit behavior consistent with real-world objects.

The overview of the proposed pipeline is illustrated in Fig. 1, which consists of three modules: a motion-factorized dynamic 3D Gaussian network (Sec. 4.2) for 4D reconstruction of the object, a coarse-to-fine density field generation strategy (Sec. 4.3) for continuum generation, surface extraction, and Gaussian attribute assignment, and a procedure (Sec. 4.4) showing how we leverage Gaussian-informed continuum and extracted surfaces to estimate physical properties.

### 4.2 Motion-factorized Dynamic 3D Gaussian Network

Our dynamic 3D Gaussian network follows existing frameworks [16, 29, 30] that simultaneously maintain a canonical 3D Gaussian set and a deformation field modeled by a neural network to warp the canonical shape into object states at specific times. The core idea of this pipeline, presented in Fig. 2, is that the motion of every point in the object can be decomposed into a small range of motion bases.

**Architecture**. We first factorize the entire motion into $N_m$ bases that are modeled by a fully connected neural network, where every basis shares a common backbone except the final layer. The output of each basis consists of the deformations at position $d\mu_i(t) \in \mathbb{R}^3$ and at scale $ds_i(t) \in \mathbb{R}$. To model the exact deformation for each position, we next propose a lightweight coefficient network that maps the positions at canonical space with specific time to their corresponding motion coefficients $w(\mu_0, t) \in \mathbb{R}^{N_m}$. Therefore, the deformed position and the scale for each Gaussian point are evaluated by the linear combination of the motion basis according to the motion coefficients:

$$\mu(t) = \mu_0 + \sum_{i=1}^{N_m} w_i(\mu_0, t)d\mu_i(t), \qquad s(t) = s_0 + \sum_{i=1}^{N_m} w_i(\mu_0, t)ds_i(t). \tag{3}$$

In this work, we regard all the Gaussians as isotropic kernels, which has been demonstrated as an effective way to simplify the model and better reconstruct the scene [6, 53]. We should note that

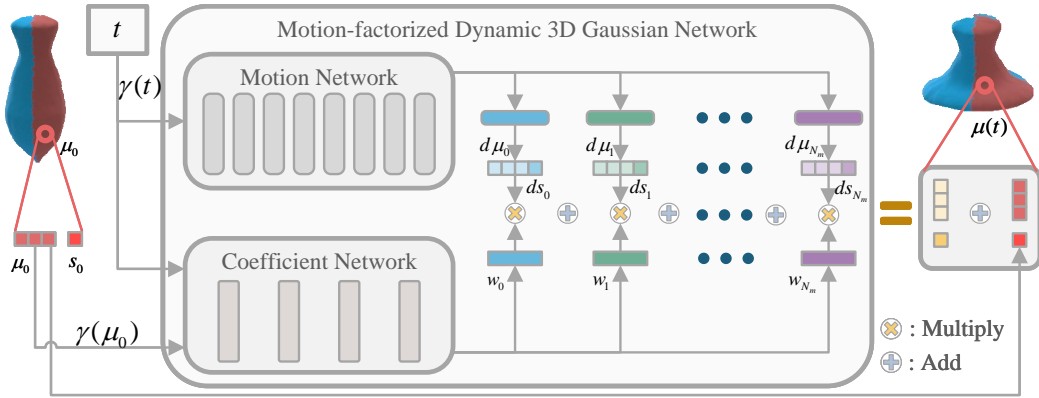

Figure 2: The pipeline of the proposed dynamic 3D Gaussian network. The motion network backbone consists of 8 fully connected (FC) layers. The output of the motion block is fed to $N_m$ heads to generate motion residuals. The coefficient network contains 4 FC layers.

although previous works [29, 54] also perform motion decomposition modeling, our pipeline shows two major differences: 1) instead of modeling each basis with an independent neural network, our module shares a common backbone. Our key observation is that for reconstructing a dynamic object, all points on the object should follow a similar moving tendency, and the final heads of the neural network are sufficient to model the details of different parts of the object; 2) to increase the ability to fit high rank of the dynamic scene [16], we model the motion coefficients as time-variant variables rather than constant Gaussian attributes [29].

**Optimization**. We employ the same setting in [16] to train our pipeline. Concretely, the canonical 3D Gaussians are initialized with points randomly sampled from the given bounding box of the scene. We start training the deformation network after 3,000 iterations of warm-up for the 3D Gaussians. Similar to previous works [16, 29], we optimize the pipeline by computing the L1 norm and Structural Similarity Index Measure (SSIM) between the rendered image $I$ and the ground truth image $\tilde{I}$. Moreover, since large scales may lead to inaccurate reconstructed shapes [55], we thus perform L1 norm on the scale attributes of all the points to recover more fine-grand shapes of the object. Therefore, the overall loss function is defined as:

$$\mathcal{L}_{gs} = \mathcal{L}_1(I, \tilde{I}) + \lambda_1 \mathcal{L}_{ssim}(I, \tilde{I}) + \lambda_2 \mathcal{L}_1(s(t)), \tag{4}$$

where $\lambda_1$ and $\lambda_2$ are balancing hyperparameters. More in-depth analysis of the proposed pipeline, including implementation details and effects of scale regularization, are presented in Appendix A.1.

### 4.3 Gaussian-informed Continuum Generation

**Coarse-to-fine density field generation**. Since the reconstructed Gaussian particles are served for rendering only, meaning that they are not evenly distributed on the objects, they cannot be directly used for simulation [51]. Therefore, we propose a novel coarse-to-fine filling strategy to iteratively generate density fields of the object based on the reconstructed Gaussian particles from Eqn. 3 and the internal particles filtered by the rendered depth maps. The proposed strategy is presented in Alg. 1. The implementation details and visual results are illustrated in Appendix A.2.

Concretely, the internal particles, initialized by uniform sampling from the bounding box of Gaussian particles, are filtered by projecting the particles to various images to compare the projected depth with rendered depth values (lines 1-6 in Alg. 1). The resulting particles can roughly represent the shape of the object. However, as denoted in Eqn. 2, the rendered depth maps are evaluated in an accumulated manner, making them less precise in representing the object surface.

Therefore, We employ a coarse-to-fine filling strategy by iteratively upsampling the density field and reassigning the densities on the indices computed from both the Gaussian and internal particles (lines 8-16 in Alg. 1). Fig. 3 provides a sketch illustration of the proposed strategy. Specifically, due to the large grid size at the initial stage, the object is completely inside the voxels with high densities. Next, we sequentially perform upsampling (line 10), mean filtering (line 13), and reassigning the

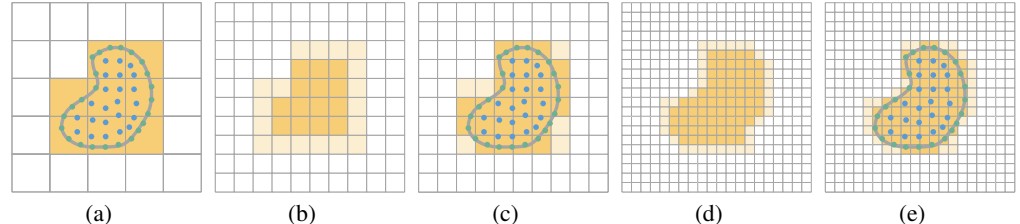

Figure 3: Sketch illustration of the coarse-to-fine filling strategy. Gaussian and internal particles are depicted in green and blue, respectively. (a) Voxels containing particles are assigned high densities. (b) Following the upsampling and smoothing of the field, densities near boundaries become blurred (indicated in light yellow). (c) The particles are again used to correct the voxels that contain particles with high densities. (d) and (e) repeat the previous operations to achieve a more detailed shape.

field (line 14) at each iteration. The first two operations produce more fine-grained shapes, and the reassigning operation ensures high densities at the surface to avoid over-erosion caused by the first two steps. Finally, the continuum particles with the corresponding object surfaces can be extracted by thresholding the density field (lines 16-17 in Alg. 1).

---

**Algorithm 1** Pseudo code for coarse-to-fine filling

---

**Input:**
    Gaussian particles at time $t$: $\mathbb{P}_G(t) = \{(\mu(t), s(t), \sigma, c)\}$;
    $n$ pairs of camera extrinsic and intrinsic parameters: $\{(T_i, K_i) | i = 1...n\}$;
    parameters: grid size $\Delta x$; number of upsampling steps $n_u$; thresholds $th_{min}, th_{min}$;
**Output:**
    Continuum particles $\tilde{P}(t)$ and the corresponding surface $\tilde{S}(t)$;
  1: Randomly sample an initial particle set $P_{in}$ from the bounding box of $\{\mu(t)\}$;
  2: **for** $i \leftarrow 1, n$ **do**
  3:    $\tilde{D}_i = GaussianSplatting(\mathbb{P}_G(t), T_i, K_i)$;            ▷ render depth map at view $i$
  4:    $(u_{in}, v_{in}), d_{in} \leftarrow Proj(P_{in}, T_i, K_i)$;      ▷ obtain image indices and depths of $P_{in}$ at view $i$
  5:    $P_{in} \leftarrow P_{in}[\tilde{D}_i(u_{in}, v_{in}) \le d_{in}]$;        ▷ filter out particles that are outside the object
  6: **end for**
  7: Initialize the zero-value density field $F(t)$ with $\Delta x$ and the bounding box of $\{\mu(t)\}$;
  8: **for** $j \leftarrow 1, n_u$ **do**
  9:    **if** $j \ne 1$ **then**
10:        $F(t) \leftarrow TrilinearInterpolation(F(t), 2)$        ▷ upsample $F(t)$ with scale factor 2
11:        $F(t)[p, q, r] = 1$, where $p, q, r \leftarrow Discretize(P_{in} \cup \{\mu(t)\})$;
12:    **end if**
13:    $F(t) \leftarrow MeanFiltering(F(t))$;
14:    $F(t)[p, q, r] = 1$, where $p, q, r \leftarrow Discretize(P_{in} \cup \{\mu(t)\})$;
15: **end for**
16: $\tilde{P}(t) \leftarrow GetPosition(th_{min} \le F(t))$;
17: $\tilde{S}(t) \leftarrow GetPosition(th_{min} \le F(t) \le th_{max})$;

---

**Gaussian-informed continuum**. In PAC-NeRF, the particles are equipped with appearance features to enable image rendering for the continuum at different states. We can also achieve this function by treating the particles as Gaussian kernels and re-train the particles using the visual data. However, this process is cumbersome and will also face the same issue in PAC-NeRF where distorted RGB images will be rendered when large deformation occurs. Therefore, instead of injecting appearance attributes, we opt to assign density and scale attributes to the particles where the densities originate from the density field, and the scale attributes can be directly obtained by the field grid size. The Gaussian-informed continuum is defined as a set of triplets:

$$\mathbb{P}_{\tilde{P}} = \{(\tilde{p}, s_{\Delta x}, \sigma_F)\}, \quad (5)$$

where $\tilde{p} \in \tilde{P}$, $s_{\Delta x} = \Delta x / 2^{n_u}$, and $\sigma_F = F[Discretize(\tilde{p})]$ (we neglect $t$ in the notation for simplicity). Therefore, we only render object masks as 2D shape surrogates for supervision.

## 4.4 Geometry-aware Physical Property Estimation

With the Gaussian-informed continuum at initial state $\mathbb{P}_{\tilde{P}}(0)$ and the extracted surfaces $\tilde{S}(t)$ in place, we can employ MPM to perform simulation on the continuum and evaluate the difference in terms of both the 3D and 2D shapes. Concretely, after a rollout by MPM given the current estimation of physical parameters, we obtain a trajectory $P(t)$ with corresponding object surfaces $S(t)$. We thus can render object masks over the trajectory. Then the loss of the current rollout can be computed as:

$$\mathcal{L}_{ppe} = \frac{1}{m}\sum_{i=1}^{m}[\mathcal{L}_{CD}(S(t_i), \tilde{S}(t_i)) + \frac{1}{n}\sum_{j=1}^{n}\mathcal{L}_1(A_j(t_i), \tilde{A}_j(t_i))], \qquad (6)$$

where $\mathcal{L}_{CD}$ and $\mathcal{L}_1$ are chamfer distance and L1 norm respectively, $S(t_i)$ denotes the simulated surface at time $t_i$, $A_j(t_i)$ is the rendered mask at view $j$, and $\tilde{A}_j(t_i)$ represents the object mask of the image extracted from video $V_j$ at time $t_i$. Due to the differential property of the simulator, the evaluated loss is used to optimize the target physical parameters $\Theta$.

## 5 Experiments

**Datasets**. To thoroughly assess our proposed method, we employ two sources of data introduced by PAC-NeRF [12] and Spring-Gaus [6]. Concretely, PAC-NeRF contributes two synthetic datasets generated by MLS-MPM framework [18]. Each object in both datasets includes RGB images from 11 distinct viewpoints, with approximately 14 frames per viewpoint. The datasets feature a range of materials, including elastic and plastic objects, granular media, and both Newtonian and non-Newtonian fluids. The first dataset contains 45 cross-shape objects with different initial conditions and ground-truth values of physical properties, while the second one consists of 9 objects with different shapes. The interpretation of the physical parameters is listed in Appendix A.9 and A.10. Spring-Gaus generates a synthetic dataset of elastic objects and collects a real-world dataset containing both static and dynamic scenes. The synthetic data contains 30 frames in each of 10 viewpoints. While the real-world data only contains 3 viewpoints for each object in the dynamic scene, it captures 50-70 images from various viewpoints for the static scene. Moreover, we follow previous works [12, 6] and use the off-the-shelf matting [56] or segmentation [57] techniques to obtain object masks.

**Baselines**. For dynamic reconstruction, we compare with PAC-NeRF and the current state-of-the-art deformable 3D Gaussian method DefGS [16] on the PAC-NeRF synthetic dataset. More comparison of our dynamic 3D Gaussian pipeline on other widely-used datasets such as D-NeRF [25] is presented in Appendix A.1.3. For system identification, we employ PAC-NeRF as the baseline and evaluate the performance using the two datasets introduced in PAC-NeRF. To further demonstrate the precision of the proposed method in terms of geometry recovery and future prediction, we perform experiments on the Spring-Gaus synthetic dataset and compare the results with PAC-NeRF and Spring-Gaus.

**Metrics**. The evaluation metrics in the experiments include 1) Chamfer Distance (CD), with units expressed in $10^3 mm^2$; 2) Earth Mover's Distance (EMD); 3) Peak Signal-to-Noise Ratio (PSNR); 4) Structural Similarity Index Metric (SSIM) [58]; and 5) Mean Absolute Error (MAE), with values scaled by a factor of $100$. The first two metrics are used to evaluate discrepancies between the reconstructed and ground-truth point clouds. PSNR and SSIM are leveraged on the Spring-Gaus dataset to validate the precision of future state prediction. We compute the mean absolute error for the evaluation of physical property estimation.

### 5.1 Evaluation on PAC-NeRF Synthetic Dataset

**Comparison on dynamic reconstruction**. In this experiment, we first perform dynamic Gaussian reconstruction on the cross-shaped object dataset using DefGS and our proposed method, respectively. We then employ the same filling strategy on the reconstructed Gaussians at each time state to generate the continuum, which is regarded as the final recovered geometry of the object and used to make comparisons with the oracle shape to compute CD and EMD. Since PAC-NeRF jointly recovers both geometries and physical parameters, we use the final estimated results to generate the trajectory for evaluation.

The results, reported in Tab. 1, show that our method outperforms the baselines on both metrics and achieves more precise reconstruction performance on most objects. Specifically, we find that

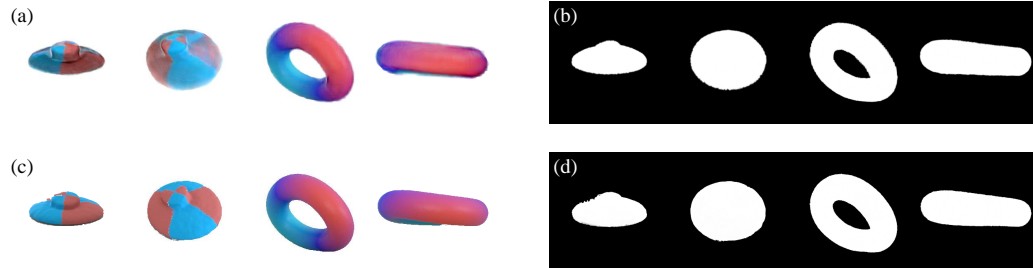

Figure 4: Comparison between rendered and ground-truth images. (a) Rendered RGB images by PAC-NeRF. (b) Rendered masks by our method. (c)-(d) Ground-truth RGB images and masks. The mask-based supervision can introduce fewer discrepancies compared with the RGB-based guidance when the estimated shapes are correct.

Table 1: Dynamic Reconstruction on PAC-NeRF Dataset

| Metrics | CD ↓ | | | EMD ↓ | | |
|---|---|---|---|---|---|---|
| Methods | PAC-NeRF [12] | DefGS [16] | Ours | PAC-NeRF [12] | DefGS [16] | Ours |
| Newtonian | 0.277 | 0.269 | **0.243** | 0.027 | 0.027 | **0.025** |
| Non-Newtonian | 0.236 | 0.216 | **0.195** | 0.025 | 0.024 | **0.022** |
| Elasticity | 0.238 | 0.191 | **0.178** | 0.025 | 0.022 | **0.02** |
| Plasticine | 0.429 | 0.213 | **0.196** | 0.029 | 0.024 | **0.022** |
| Sand | **0.212** | 0.281 | 0.25 | 0.025 | 0.028 | **0.025** |
| Mean | 0.278 | 0.234 | **0.212** | 0.026 | 0.025 | **0.023** |

the NeRF representation used by PAC-NeRF usually leads to overly large shape generation. While DefGS performs well on elastic objects, its performance degenerates when modeling objects with large deformations, such as granular media and fluids. Our method can better handle these objects due to the flexibility of trajectory representation.

**Comparison on system identification.** We evaluate the performance of system identification of the two datasets proposed by PAC-NeRF. For the first dataset, we compute the MAE of the parameters for each type of object. To demonstrate the effectiveness of the 2D shape representation, we also conduct experiments on the second dataset by only using masks for supervision on our method, namely "Ours*". For the second dataset, we execute 10 times of our method with different random seeds for each object instance and report the mean value of the estimation results. The training details are illustrated in Appendix A.3.

The results, reported in Tab. 2 and Tab. 3, show that the proposed hybrid pipeline can achieve more accurate estimation over a wide range of entries and objects, which demonstrate the effectiveness of the geometry-aware guidance. Fig. 4 visualizes the RGB images rendered by PAC-NeRF and the masks rendered by our method. We can see that when large deformation occurs, the rendered RGB image becomes distorted, while the rendered mask can effectively reduce such effect and get better perfor-

Table 2: System identification performance on PAC-NeRF cross-shaped object Dataset

| Type | Parameters | PAC-NeRF | Ours* | Ours |
|---|---|---|---|---|
| Newtonian | $\log_{10}(\mu)$ | 11.6±6.60 | 1.53±1.45 | **1.53**±1.31 |
| | $\log_{10}(\kappa)$ | 16.7±5.37 | 16.0±22.4 | **14.8**±19.2 |
| | $v$ | 0.86±1.45 | 0.20±0.08 | **0.20**±0.07 |
| Non-Newtonian | $\log_{10}(\mu)$ | 24.1±21.9 | 32.9±44.6 | **13.5**±18.2 |
| | $\log_{10}(\kappa)$ | 44.0±26.3 | 17.7±20.2 | **12.9**±16.8 |
| | $\log_{10}(\tau_Y)$ | 5.09±7.41 | **3.74**±3.72 | 4.80±3.92 |
| | $\log_{10}(\eta)$ | **28.7**±23.3 | 34.9±24.1 | 40.7±24.6 |
| | $v$ | 0.29±0.13 | 0.68±0.28 | **0.19**±0.09 |
| Elasticity | $\log_{10}(E)$ | 3.02±3.72 | 3.27±4.13 | **2.43**±3.29 |
| | $\nu$ | 4.35±5.08 | 3.10±2.00 | **2.52**±2.03 |
| | $v$ | **0.50**±0.23 | 0.78±0.26 | 0.82±0.32 |
| Plasticine | $\log_{10}(E)$ | 83.8±68.4 | 28.1±24.4 | **25.6**±29.4 |
| | $\log_{10}(\tau_Y)$ | 11.2±14.5 | **1.24**±0.90 | 1.67±1.21 |
| | $\nu$ | 18.9±15.7 | 10.2±5.34 | **9.59**±5.00 |
| | $v$ | 0.56±0.17 | **0.13**±0.04 | 0.22±0.10 |
| Sand | $\theta_{fric}$ | 4.89±1.10 | 4.21±0.08 | **4.18**±0.52 |
| | $v$ | 0.21±0.08 | 0.24±0.08 | **0.17**±0.05 |

Table 3: System Identification Performance on PAC-NeRF Dataset

| | PAC-NeRF [12] | Ours | Ground Truth |
|---|---|---|---|
| Droplet | $\mu = 2.09 \times 10^2, \kappa = \mathbf{1.08 \times 10^5}$ | $\mu = \mathbf{2.01 \times 10^2}, \kappa = 0.18 \times 10^5$ | $\mu = 200, \kappa = 10^5$ |
| Letter | $\mu = 83.85, \kappa = 1.35 \times 10^5$ | $\mu = \mathbf{95.05}, \kappa = \mathbf{1.00 \times 10^5}$ | $\mu = 100, \kappa = 10^5$ |
| Cream | $\mu = 1.21 \times 10^5, \kappa = 1.57 \times 10^6$, $\tau_Y = 3.16 \times 10^3, \eta = 5.6$ | $\mu = \mathbf{1.03 \times 10^4}, \kappa = \mathbf{1.48 \times 10^6}$, $\tau_Y = \mathbf{2.98 \times 10^3}, \eta = 6.6$ | $\mu = 10^4, \kappa = 10^6$, $\tau_Y = 3 \times 10^3, \eta = 10$ |
| Toothpaste | $\mu = 6.51 \times 10^3, \kappa = 2.22 \times 10^5$, $\tau_Y = 228, \eta = \mathbf{9.77}$ | $\mu = \mathbf{4.19 \times 10^3}, \kappa = \mathbf{9.24 \times 10^4}$, $\tau_Y = \mathbf{226}, \eta = 9.1$ | $\mu = 5 \times 10^3, \kappa = 10^5$, $\tau_Y = 200, \eta = 10$ |
| Torus | $E = 1.04 \times 10^6, \nu = 0.322$ | $E = \mathbf{0.99 \times 10^6}, \nu = \mathbf{0.295}$ | $E = 10^6, \nu = 0.3$ |
| Bird | $E = 2.78 \times 10^5, \nu = 0.273$ | $E = \mathbf{3.08 \times 10^5}, \nu = \mathbf{0.284}$ | $E = 3 \times 10^5, \nu = 0.3$ |
| Playdoh | $E = 3.84 \times 10^6, \nu = 0.272, \tau_Y = 1.69 \times 10^4$ | $E = \mathbf{1.58 \times 10^6}, \nu = \mathbf{0.322}, \tau_Y = \mathbf{1.56 \times 10^4}$ | $E = 2 \times 10^6, \nu = 0.3, \tau_Y = 1.54 \times 10^4$ |
| Cat | $E = 1.61 \times 10^5, \nu = 0.293, \tau_Y = 3.57 \times 10^3$ | $E = \mathbf{0.98 \times 10^6}, \nu = \mathbf{0.296}, \tau_Y = \mathbf{3.76 \times 10^3}$ | $E = 10^6, \nu = 0.3, \tau_Y = 3.85 \times 10^3$ |
| Trophy | $\theta_{fric}^0 = 36.1°$ | $\theta_{fric}^0 = \mathbf{38.0°}$ | $\theta_{fric}^0 = 40°$ |

Table 4: Future State Simulation on Spring-Gaus Synthetic Dataset

| | | torus | cross | cream | apple | paste | chess | banana | Mean |
|---|---|---|---|---|---|---|---|---|---|
| CD$\rightarrow$ | Spring-Gaus [6] | 2.38 | 1.57 | 2.22 | 1.87 | 7.03 | 2.59 | 18.48 | 5.16 |
| | PAC-NeRF [12] | 2.47 | 3.87 | 2.21 | 4.69 | 37.7 | 8.2 | 66.43 | 17.94 |
| | Ours | **0.75** | **1.09** | **0.94** | **0.22** | **2.79** | **0.77** | **0.12** | **0.95** |
| EMD$\rightarrow$ | Spring-Gaus [6] | 0.087 | **0.051** | 0.094 | 0.076 | 0.126 | 0.095 | 0.135 | 0.095 |
| | PAC-NeRF [12] | 0.055 | 0.111 | 0.083 | 0.108 | 0.192 | 0.155 | 0.234 | 0.134 |
| | Ours | **0.034** | 0.058 | **0.050** | **0.030** | **0.096** | **0.059** | **0.017** | **0.049** |
| PSNR$\uparrow$ | Spring-Gaus [6] | 16.83 | 16.93 | 15.42 | 21.55 | 14.71 | 16.08 | 17.89 | 17.06 |
| | PAC-NeRF [12] | 17.46 | 14.15 | 15.37 | 19.94 | 12.32 | 15.08 | 16.04 | 15.77 |
| | Ours | **20.24** | **30.51** | **19.15** | **26.89** | **16.31** | **18.44** | **29.29** | **22.98** |
| SSIM$\uparrow$ | Spring-Gaus [6] | 0.919 | **0.940** | 0.862 | 0.902 | 0.872 | 0.881 | 0.904 | 0.897 |
| | PAC-NeRF [12] | 0.913 | 0.906 | 0.858 | 0.878 | 0.819 | 0.848 | 0.886 | 0.870 |
| | Ours | **0.942** | 0.939 | **0.909** | **0.948** | **0.894** | **0.912** | **0.964** | **0.930** |

mance. By leveraging both 3D and 2D shape guidance, our method obtains the best results on most entries. More qualitative results are available in the supplementary video.

## 5.2 Evaluation on Spring-Gaus Synthetic Dataset

**Comparison on future state simulation**. To further demonstrate the performance of our proposed method, we follow the setting in Spring-Gaus [6] that uses the first 20 frames as training data and the subsequent 10 frames for evaluation. Concretely, we first perform system identification based on our method and then use the estimated physical parameters and the continuum to simulate a trajectory that includes the states of the 30 frames. Therefore, we can compute CD and EMD between the simulated continuum and the ground-truth point cloud. Since we know the exact position of the continuum at each time state after estimation, we can assign time-invariant Gaussian attributes by training Gaussians on the continuum using the first 20 frames of RGB images, which enable image rendering at novel views and states. Therefore, we can compute PSNR and SSIM at any time state.

The results of future state prediction are presented in Tab. 4, and the results of reconstruction on the training states are reported in Appendix A.4. We observe that our method significantly outperforms the baselines on CD and EMD metrics over almost all object instances, which shows the superiority of our method for both geometry recovery and system identification. The results of PSNR and SSIM show that leveraging dynamic visual data to train the Gaussian attributes on the continuum improves rendering quality. This further reveals that the generated trajectories are precise such that the particles are consistent to contribute to the rendering for the same region of the object at different time states.

## 5.3 Real-world Application: Digital Twins in Robotic Grasping Scenario

To demonstrate the efficacy of the proposed method in real-world scenarios, we perform system identification on the real-world dataset collected by Spring-Gaus [6], as shown in Fig. 5. Since the real-world dataset consists of static and dynamic scenes for each object, we follow the procedure introduced by Spring-Gaus to progressively 1) reconstruct a Gaussian set of the object from the

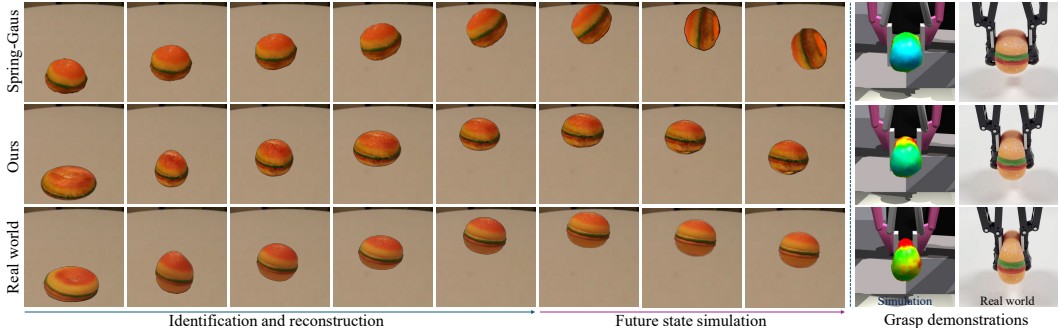

Figure 5: Real-world application. Left: Identification and future state simulation. Right: Grasping simulation. The stress on the simulated object is indicated by blue (low) to red (high). The gripper widths from top to bottom are set to 6cm, 4.5cm, and 3.5cm, respectively.

static scene, 2) transform the static Gaussian set to the initial state of the dynamic scene based on a registration network similar as iNeRF [6, 59], and 3) perform system identification from the dynamic observation by our method "Ours*" due to the lack of sufficient images for dynamic reconstruction. Subsequently, we establish robotic platforms in both simulated and real-world environments, each equipped with UR10 robot arms configured identically. We then execute grasp attempts on both the reconstructed objects with the estimated properties in the simulation and the corresponding real-world objects under the same configuration. The results of more objects, and more details about the training and the experiment setting are presented in Appendix A.5. From the results shown in Fig. 5, we see that our method demonstrates its capability to effectively model the deformation experienced by the objects upon impact with a surface. Furthermore, by applying identical gripper forces to both the simulated and real-world versions of the objects, we observe similar deformation behaviors. This consistency in deformation under identical conditions supports that the estimated physical parameters closely mirror the real-world properties of the objects.

## 6  Conclusion and Limitations

This paper proposes a novel solution that leverages the 3D Gaussian representation of objects to acquire explicit shapes while concurrently enabling the simulated continuum to render 2D shapes to facilitate the estimation of physical properties. A novel motion-factorized dynamic 3D Gaussian framework is proposed to reconstruct precise dynamic scenes. Object surfaces and Gaussian-informed continuum are obtained by utilizing the proposed coarse-to-fine density field generation strategy. Extensive experiments demonstrate the efficacy and applicability of our method.

Despite the performance we achieve, this method still suffers from limitations, such as the assumption of continuum mechanics, the requirements of multi-view images with known camera poses, and the need for prior knowledge of object constitutive models. Integrating the pose-free method [60] or generalized constitutive [61] model with our method will be an interesting direction for future work.

From the perspective of application, while this method can yield accurate estimations, it may pose risks for fragile objects, as the interaction required for property inference could potentially cause damage. Moreover, the computational demands of our framework are substantial which require at least 1.5 hours to simultaneously recover both the geometry and physical properties of each object. Future work could explore leveraging multi-model large language models [62] and large reconstruction models [63–66] to facilitate the recovery process.

## 7  Acknowledgements

This research was supported by the Research Grant Council of the Hong Kong Special Administrative Region under grant number 16212623. We thank Licheng Zhong for providing us with details about real data collection and links for purchasing objects for real-world experiments.

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

# A   Appendix

## A.1   Motion-factorized Dynamic 3D Gaussian Network

### A.1.1   Implementation details

We employ temporal and positional encoding to the time $t$ and position $\mu_0$, respectively, to introduce features with various frequencies. Specifically, the encoding module is denoted as $\gamma(x) = \left(\sin(2^k\pi x), \cos(2^k\pi x)\right)_{k=0}^{L-1}$, where $L = 10$ for both $t$ and $\mu_0$.

All the modules within the proposed network are composed of fully connected layers. The intermediate layers are uniformly designed, featuring both input and output channels configured to 256, and employ ReLU activation. For training, we adhere to the protocol established in [16], utilizing the Adam optimizer [67] with the same learning rate as specified in [16]. The total number of iterations is set at 40,000, with densification and pruning operations conducted every 500 steps until reaching 15,000 iterations. Additionally, the number of motions $N_m$ is set to 8 for all objects in our network. $\lambda_1$ and $\lambda_2$ in Eqn. 4 are all set to 1. All the experiments are conducted on a single A10 GPU.

### A.1.2   Effects of scale regularization

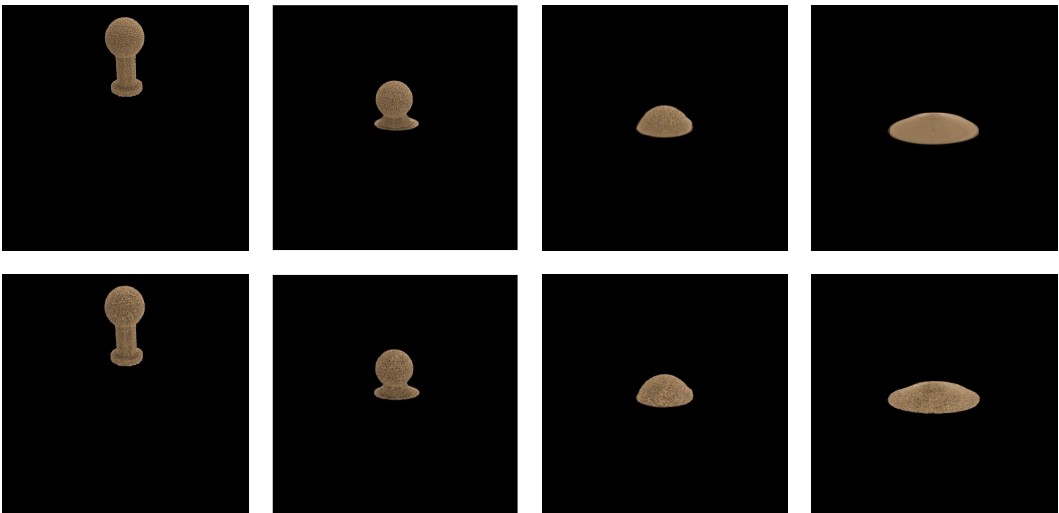

Figure 6: Visualization of trophy sequences. Row 1: rendering results from the network trained without scale regularization. Row 1: rendering results from the network trained with scale regularization.

When addressing the deformation of objects such as fluids or granular media, the network may struggle to fit transformations accurately due to significant discrepancies between the canonical and target shapes. As a compensatory mechanism, the network may employ Gaussians with enlarged scales to mitigate shape distortions during image rendering. This effect is visualized in the top row of Fig. 6. To rectify this issue, we implement scale regularization during network training, which enforces Gaussian kernels to maintain smaller scales. The efficacy of this operation is demonstrated in the second row of Fig. 6, where it is evident that scale regularization enables the reconstruction of more precise shapes for rendering.

### A.1.3   Evaluation on D-NeRF Dataset

To further evaluate the performance of our method in terms of novel view synthesis, we conduct the experiment on the D-NeRF [25] dataset, which is a widely used benchmark consisting of moving items with data captured by a monocular camera. We compute PSNR on the D-NeRF test set and compare our method with previous dynamic approaches, including Tensor4D [68], K-Planes [69], TiNeuVox [70], and DefGS [16]. The results, reported in Tab. 5, demonstrate the proposed dynamic 3D Gaussian pipeline can also achieve superior performance on rendering.

Table 5: Results of PSNR ($\uparrow$) on D-NeRF [25] Dataset

| Method | Hell Warrior | Mutant | Hook | Bouncing Balls | T-Rex | Stand Up | Jumping Jacks | Mean |
|---|---|---|---|---|---|---|---|---|
| Tensor4D [68] | 31.26 | 29.11 | 28.63 | 24.47 | 23.86 | 30.56 | 24.2 | 27.44 |
| K-Planes [69] | 24.58 | 32.5 | 28.12 | 40.05 | 30.43 | 33.1 | 31.11 | 31.41 |
| TiNeuVox [70] | 27.1 | 31.87 | 30.61 | 40.23 | 31.25 | 34.61 | 33.49 | 32.74 |
| DefGS [16] | 41.54 | 42.63 | 37.42 | 41.01 | **38.1** | 44.62 | 37.72 | 40.43 |
| Ours | **41.97** | **42.93** | **38.04** | **41.26** | 37.54 | **45.32** | **38.86** | **40.85** |

## A.2 Gaussian-informed Continnum Generation

### A.2.1 Implementation details

In Alg. 1, the number of iterations, denoted as $n_u$, is uniformly set to 4 for all objects. We set the initial grid size $\Delta x$ according to the volume of the object. For most objects, $\Delta x = 0.1$, while for small items such as toothpaste in PAC-NeRF dataset, $\Delta x = 0.01$. The parameters $th_{min}$ and $th_{max}$ are set to 0.5 and 0.8, respectively. The resulting particle count ranges from approximately 50,000 to 100,000.

### A.2.2 Visualization of coarse-to-fine filling

Fig. 7 visualizes the filling results of our proposed coarse-to-fine strategy with different numbers of iterations, along with the results from PAC-NeRF and ground-truth shapes. The qualitative results show that our method can generate more accurate shapes compared with PAC-NeRF, which tends to recover over-large shapes. We should note that we cannot recover the cat-shaped object as in [12], though we use the code officially implemented by PAC-NeRF without any modification.

## A.3 Training details on PAC-NeRF Dataset

The training process is divided into two sub-processes, where we perform system identification after estimating the initial velocity of the object using the first three frames of data. Both processes use Adam [67] optimizer to tune the parameters.

## A.4 More Experiments on Spring-Gaus Synthetic Dataset

Besides performing evaluation on the simulated future states in Sec. 5.2, we also evaluate CD and EMD on states existing in the training data, and the results are reported in Tab. 6. It is obvious to see that our method outperforms the baselines by a large margin, which further demonstrates the performance of our method in terms of reconstruction and identification.

Table 6: Dynamic Reconstruction on Spring-Gaus Synthetic Dataset

| | | torus | cross | cream | apple | paste | chess | banana | Mean |
|---|---|---|---|---|---|---|---|---|---|
| CD$\downarrow$ | Spring-Gaus [6] | 0.17 | 0.48 | 0.36 | 0.38 | 0.19 | 1.80 | 2.60 | 0.85 |
| | PAC-NeRF [12] | 4.92 | 1.10 | 0.77 | 1.11 | 3.14 | 0.96 | 2.77 | 2.11 |
| | Ours | **0.13** | **0.13** | **0.14** | **0.15** | **0.17** | **0.41** | **0.03** | **0.17** |
| EMD$\downarrow$ | Spring-Gaus [6] | 0.040 | 0.037 | 0.031 | 0.033 | **0.022** | 0.063 | 0.052 | 0.040 |
| | PAC-NeRF [12] | 0.056 | 0.052 | 0.041 | 0.045 | 0.054 | 0.052 | 0.062 | 0.052 |
| | Ours | **0.020** | **0.020** | **0.019** | **0.020** | 0.025 | **0.036** | **0.011** | **0.022** |

## A.5 Experiment Setting for Spring-Gaus Real-world Dataset

**Training details**. The dynamic scenes in Spring-Gaus [6] contain only three viewpoints, which are insufficient for dynamic 3D Gaussian reconstruction. Conversely, the static scenes incorporate 50 to 70 images captured from various viewpoints. Following the protocol established in Spring-Gaus, we reconstruct 3D Gaussian points from the static scenes using the traditional 3D Gaussian Splatting

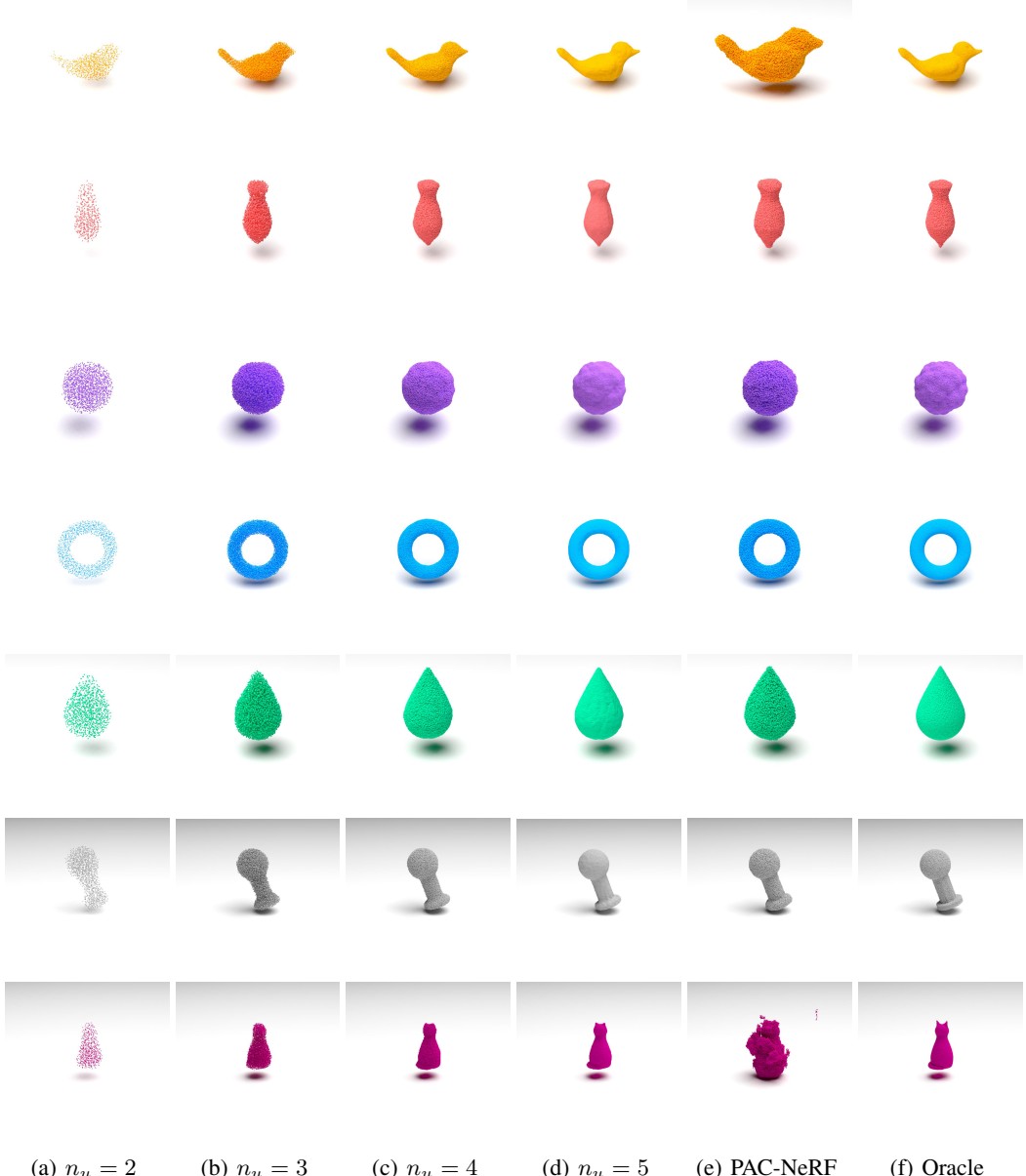

(a) $n_u = 2$    (b) $n_u = 3$    (c) $n_u = 4$    (d) $n_u = 5$    (e) PAC-NeRF    (f) Oracle

Figure 7: Visualization of Coarse-to-fine Filling. (a)-(d) are the filling results by our method with different times of upsampling operations. (e) visualize the point clouds recovered by PAC-NeRF. (f) shows the ground-truth shapes.

(3DGS) technique [15]. Subsequently, we transform the static Gaussian set to the initial configuration of the dynamic scene, guided by the relative pose between the two scenes. The pose is estimated iteratively based on the discrepancies observed between the rendered images and the actual images at the initial state of the dynamic scene. After pose estimation, we implement our methodology, which leverages only implicit shape guidance, to conduct system identification.

**Experimental setting**. We conducted grasping experiments using the UR10 robotic arm equipped with the Robotiq140 dexterous gripper in both simulated and real-world settings, ensuring consistency in the mass of the objects and their grasping poses across both environments. For the simulations, we employed the FEM-based Isaac Gym simulator [71] for its advanced capabilities in realistically simulating deformable objects [72]. To facilitate the simulation of deformable objects, we apply the Marching Cubes algorithm [73] to the generated density fields to derive the object meshes. Subsequently, we utilize fTetWild [74] for the tetrahedralization of these meshes.

**More results**. Qualitative results of grasp demonstrations on pig and dog objects are shown in Fig. 8.

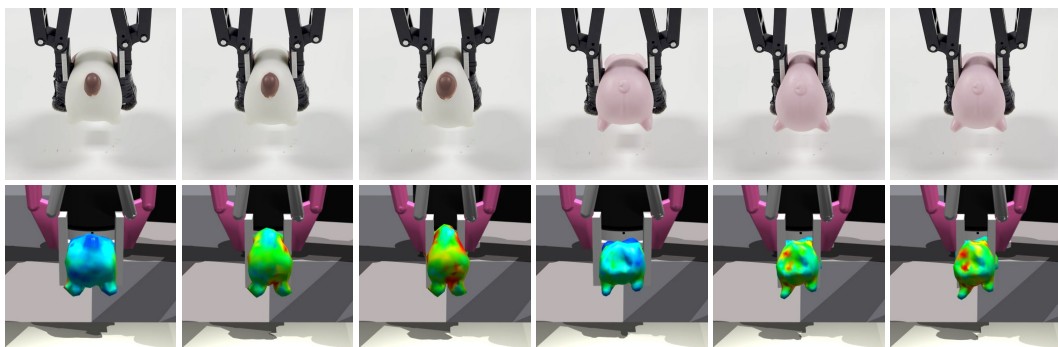

Figure 8: The stress on the simulated objects is indicated by blue (low) to red (high). The gripper widths from left to right for each object are set to 5.5cm, 4.5cm, and 3.5cm, respectively.

### A.6 System Identification Result on PAC-NeRF Dataset

Table 7: System Identification Result on PAC-NeRF Dataset

| | | | | | | | | | | |
|---|---|---|---|---|---|---|---|---|---|---|
| Newtonian (Init. Guess: $\mu = 10, \kappa = 1.0 \times 10^4$) | | | | | | | | | | |
| Prediction | $\mu = 17.43$ $\kappa = 218085.89$ | $\mu = 449.45$ $\kappa = 148550.15$ | $\mu = 160.74$ $\kappa = 177748.93$ | $\mu = 123.07$ $\kappa = 223551.76$ | $\mu = 48.48$ $\kappa = 483821.18$ | $\mu = 40.57$ $\kappa = 14668.64$ | $\mu = 68.24$ $\kappa = 336795.04$ | $\mu = 230.46$ $\kappa = 30713.74$ | $\mu = 563.53$ $\kappa = 24459.41$ | $\mu = 104.92$ $\kappa = 139609.16$ |
| Ground Truth | $\mu = 19.46$ $\kappa = 56075.55$ | $\mu = 436.62$ $\kappa = 152696.25$ | $\mu = 155.83$ $\kappa = 193525.59$ | $\mu = 121.76$ $\kappa = 257356.05$ | $\mu = 49.09$ $\kappa = 518012.47$ | $\mu = 38.44$ $\kappa = 13772.52$ | $\mu = 64.16$ $\kappa = 358237.13$ | $\mu = 228.71$ $\kappa = 11041.06$ | $\mu = 552.98$ $\kappa = 16789.77$ | $\mu = 106.93$ $\kappa = 112569.73$ |
| Non-newtonian (Init. Guess: $\mu = 100.0, \kappa = 1.0 \times 10^5, \tau_Y = 10, \eta = 1$) | | | | | | | | | | |
| Prediction | $\mu = 13201.66$ $\kappa = 218085.89$ $\tau_Y = 1246.56$ $\eta = 4.34$ | $\mu = 67041.05$ $\kappa = 148550.15$ $\tau_Y = 7679.39$ $\eta = 24.21$ | $\mu = 40092.22$ $\kappa = 177748.93$ $\tau_Y = 4238.86$ $\eta = 12.85$ | $\mu = 35080.68$ $\kappa = 223551.76$ $\tau_Y = 5289.52$ $\eta = 5.99$ | $\mu = 22017.29$ $\kappa = 483821.18$ $\tau_Y = 1037.84$ $\eta = 70.84$ | $\mu = 20142.84$ $\kappa = 14668.64$ $\tau_Y = 3930.36$ $\eta = 48.23$ | $\mu = 26122.95$ $\kappa = 336795.04$ $\tau_Y = 3604.00$ $\eta = 5.19$ | $\mu = 48006.58$ $\kappa = 30713.74$ $\tau_Y = 1441.98$ $\eta = 1.22$ | $\mu = 75068.35$ $\kappa = 24459.41$ $\tau_Y = 1422.56$ $\eta = 4.07$ | $\mu = 32390.85$ $\kappa = 139609.16$ $\tau_Y = 4960.55$ $\eta = 6.26$ |
| Ground Truth | $\mu = 13209.25$ $\kappa = 201566.59$ $\tau_Y = 1151.42$ $\eta = 6.68$ | $\mu = 65351.08$ $\kappa = 171054.03$ $\tau_Y = 7491.70$ $\eta = 26.69$ | $\mu = 43757.04$ $\kappa = 249639.94$ $\tau_Y = 3964.94$ $\eta = 23.27$ | $\mu = 36027.61$ $\kappa = 134751.55$ $\tau_Y = 5061.12$ $\eta = 22.31$ | $\mu = 19593.71$ $\kappa = 121836.33$ $\tau_Y = 1462.78$ $\eta = 38.83$ | $\mu = 20522.72$ $\kappa = 14494.30$ $\tau_Y = 4153.38$ $\eta = 27.24$ | $\mu = 51549.45$ $\kappa = 370317.66$ $\tau_Y = 3203.67$ $\eta = 20.43$ | $\mu = 121865.90$ $\kappa = 32859.59$ $\tau_Y = 1192.76$ $\eta = 10.27$ | $\mu = 241579.97$ $\kappa = 30324.98$ $\tau_Y = 1251.29$ $\eta = 10.62$ | $\mu = 33764.59$ $\kappa = 122896.10$ $\tau_Y = 4689.16$ $\eta = 22.89$ |
| Elasticity (Init. Guess: $E = 316227.77, \nu = 0.25$) | | | | | | | | | | |
| Prediction | $E = 23008.38$ $\nu = 0.3008$ | $E = 3013161.44$ $\nu = 0.2758$ | $E = 644436.88$ $\nu = 0.2875$ | $E = 431721.84$ $\nu = 0.3024$ | $E = 106731.23$ $\nu = 0.3527$ | $E = 81726.40$ $\nu = 0.1250$ | $E = 178265.23$ $\nu = 0.3291$ | $E = 1106374.86$ $\nu = 0.1731$ | $E = 4230293.75$ $\nu = 0.1583$ | $E = 339834.14$ $\nu = 0.2717$ |
| Ground Truth | $E = 21905.91$ $\nu = 0.2969$ | $E = 3844263.25$ $\nu = 0.2892$ | $E = 648705.38$ $\nu = 0.3066$ | $E = 459804.63$ $\nu = 0.2476$ | $E = 103565.68$ $\nu = 0.2954$ | $E = 81467.34$ $\nu = 0.1798$ | $E = 179237.53$ $\nu = 0.3456$ | $E = 1117086.13$ $\nu = 0.1829$ | $E = 3650610.75$ $\nu = 0.1684$ | $E = 340267.00$ $\nu = 0.2597$ |
| Plasticine (Init. Guess: $E = 10000, \nu = 0.25, \tau_Y = 1000$) | | | | | | | | | | |
| Prediction | $E = 174283.92$ $\nu = 0.3008$ $\tau_Y = 1553.91$ | $E = 4494502.24$ $\nu = 0.2758$ $\tau_Y = 58973.06$ | $E = 1607386.28$ $\nu = 0.2875$ $\tau_Y = 17967.93$ | $E = 1230654.13$ $\nu = 0.3024$ $\tau_Y = 27979.03$ | $E = 484760.99$ $\nu = 0.3527$ $\tau_Y = 1077.11$ | $E = 405734.16$ $\nu = 0.1250$ $\tau_Y = 15447.70$ | $E = 682408.49$ $\nu = 0.3291$ $\tau_Y = 12988.84$ | $E = 2304631.70$ $\nu = 0.1731$ $\tau_Y = 2079.31$ | $E = 5635256.43$ $\nu = 0.1583$ $\tau_Y = 2023.68$ | $E = 1049167.13$ $\nu = 0.2717$ $\tau_Y = 24607.02$ |
| Ground Truth | $E = 219037.98$ $\nu = 0.2620$ $\tau_Y = 1521.82$ | $E = 4974443.00$ $\nu = 0.1410$ $\tau_Y = 64508.23$ | $E = 3795741.25$ $\nu = 0.1842$ $\tau_Y = 16983.20$ | $E = 1661460.38$ $\nu = 0.1078$ $\tau_Y = 28502.29$ | $E = 456356.41$ $\nu = 0.2499$ $\tau_Y = 1070.41$ | $E = 427719.28$ $\nu = 0.1566$ $\tau_Y = 14766.65$ | $E = 5106352.50$ $\nu = 0.1743$ $\tau_Y = 12059.57$ | $E = 1382011.00$ $\nu = 0.2312$ $\tau_Y = 2050.87$ | $E = 1045986.31$ $\nu = 0.2258$ $\tau_Y = 2131.80$ | $E = 1144808.88$ $\nu = 0.1988$ $\tau_Y = 24397.29$ |
| Sand (Init. Guess: $\theta^0_{fric} = 10$) | | | | | | | | | | |
| Prediction | $\theta^0_{fric} = 32.9184$ | | $\theta^0_{fric} = 34.4715$ | | $\theta^0_{fric} = 29.1305$ | | $\theta^0_{fric} = 31.7486$ | | $\theta^0_{fric} = 45.2390$ | |
| Ground Truth | $\theta^0_{fric} = 30.6577$ | | $\theta^0_{fric} = 32.3751$ | | $\theta^0_{fric} = 26.8816$ | | $\theta^0_{fric} = 29.3458$ | | $\theta^0_{fric} = 42.2861$ | |

Each material contains 10 instances (5 for granular material) with various object orientations, initial velocities, and physical parameters.

Table 8: Notation of Algorithm 1

| Operator or symbol | Explanation |
|---|---|
| $\mathbb{P}_G(t)$ | Gaussian particle set at time $t$ |
| $\tilde{P}(t)$ | Sampled continuum particles at time $t$ |
| $\tilde{S}(t)$ | Sampled surface particles at time $t$, $\tilde{S}(t) \subset \tilde{P}(t)$ |
| $F(t)$ | 3D Density field at time $t$ |
| $Proj$ | Operation projecting 3D particles into 2D image indices according to the camera parameters |
| $Discretize$ | Operation mapping particle positions to voxel indices on the density field |
| $GetPosition$ | Operation returning 3D positions of the binary field |

## A.7 Notation of Algorithm 1

## A.8 Necessity of 2D mask supervision

To evaluate the necessity of 2D mask supervision, we perform system identification on 45 cross-shaped object instances in the PAC-NeRF dataset by our method but with only object surface supervision. The results are reported in Tab. 9. It is obvious to see that combining both 2D and 3D shapes as supervision can achieve more accurate performance compared to using 3D shapes only. Therefore, we believe that utilizing 2D mask supervision to some extent makes up for the errors introduced by the 3D object surfaces extracted from dynamic 3D Gaussians.

Table 9: System identification with/without mask supervision

| Type | Parameters | w/o masks | w/ masks |
|---|---|---|---|
| Newtonian | $\log_{10}(\mu)$ | 2.19±2.90 | **1.53±1.31** |
| | $\log_{10}(\kappa)$ | 24.2±22.2 | **14.8±19.2** |
| | $v$ | 0.20±0.08 | **0.20±0.07** |
| Non-Newtonian | $\log_{10}(\mu)$ | 19.4±27.7 | **13.5±18.2** |
| | $\log_{10}(\kappa)$ | 24.0±24.8 | **12.9±16.8** |
| | $\log_{10}(\tau_Y)$ | **4.58±9.11** | 4.80±3.92 |
| | $\log_{10}(\eta)$ | 49.1±40.5 | **40.7±24.6** |
| | $v$ | 1.33±0.54 | **0.19±0.09** |
| Elasticity | $\log_{10}(E)$ | 2.85±1.94 | **2.43±3.29** |
| | $\nu$ | 3.97±2.64 | **2.52±2.03** |
| | $v$ | **0.22±0.10** | 0.82±0.32 |
| Plasticine | $\log_{10}(E)$ | **25.6±27.4** | 25.6±29.4 |
| | $\log_{10}(\tau_Y)$ | 9.04±2.37 | **1.67±1.21** |
| | $v$ | 1.16±0.00 | **0.22±0.10** |
| Sand | $\theta_{fric}$ | **2.55±2.03** | 4.18±0.52 |
| | $v$ | 0.31±0.18 | **0.17±0.05** |

## A.9 Physical Properties

In this work, we simulate five types of materials, including elasticity, plasticine, granular media, Newtonian fluids, and non-Newtonian fluids. Each material exhibits distinct physical properties. We provide a brief introduction to the properties of each material.

*Elasticity*: The Young's modulus ($E$) is a measure of the stiffness of a solid material, quantifying the relationship between stress and strain in a material under elastic deformation. The Poisson's ratio ($\nu$) describes the tendency of a material to expand or contract along its width when it is stretched or compressed along its length.

*Plasticine*: The yield stress ($\tau_Y$) is the minimum stress that a material requires to transition from elastic deformation to plastic deformation, marking the onset of permanent deformation. Both Young's modulus ($E$) and Poisson's ratio ($\nu$) exhibit characteristics similar to those of elastic materials.

*Granular Media*: The friction angle ($\theta_{fric}$) is a measure of the inherent resistance of a granular material to sliding or shearing, directly related to the angle at which a material can be piled without slumping.

*Newtonian fluids*: The bulk modulus ($\kappa$) is a measure of a material's resistance to uniform compression, quantifying how much it compresses under a given amount of external pressure. Fluid viscosity ($\mu$) describes a fluid's resistance to flow, quantifying how much it resists deformation at a given rate.

*Non-Newtonian fluids*: The plasticity viscosity ($\eta$) refers to the measure of a viscoplastic material's resistance to deformation, which defines how it behaves under stress beyond its yield point. The bulk modulus ($\kappa$) and fluid viscosity ($\mu$) are comparable to those of Newtonian fluids, while the yield stress ($\tau_Y$) is akin to that of plasticine.

### A.10 Constitutive Models

A constitutive model describes how a material responds to stress, strain, or other external forces. It defines the material's behavior by relating stress and strain through constitutive equations, which can capture complex behaviors such as elasticity, plasticity, and fracture. The MPM simulator is capable of modeling a diverse range of materials by employing various constitutive models. In this work, we have implemented simulations for five distinct types of materials: elasticity, plasticine, granular, Newtonian fluids, and non-Newtonian fluids.

**Elasticity**. We use the Neo-Hookean model, which is a common nonlinear hyperelastic model, to simulate the elasticity of materials and predict deformations. The Cauchy stress for this model is defined by

$$J\boldsymbol{\sigma} = \mu\left(\mathbf{F}\mathbf{F}^\mathsf{T}\right) + \left[\lambda\log(J) - \mu\right]\mathbf{I}, \tag{7}$$

where the $\mathbf{F}$ is the deformation gradient, $J = \det(\mathbf{F})$ and $\mu, \lambda$ are the Lamé parameters, which are related to the material properties of Young's modulus ($E$) and Poisson's ratio ($\nu$) as:

$$\mu = \frac{E}{2(1+\nu)}, \qquad \lambda = \frac{E\nu}{(1+\nu)(1-2\nu)}. \tag{8}$$

**Plasticine**. We use the Saint Venant-Kirchhoff Model (StVK) together with von Mises yield criterion to simulate the plasticine. For this model, the stess is defined as:

$$J\boldsymbol{\sigma} = \mathbf{F}\left[2\mu\mathbf{G} + \lambda\mathrm{Tr}(\mathbf{G})\mathbf{I}\right]\mathbf{F}^\mathsf{T}, \tag{9}$$

where $\mathbf{G} = \frac{1}{2}\left(\mathbf{F}^\mathsf{T}\mathbf{F} - \mathbf{I}\right)$ is the Green strain. The von Mises yield criterion serves as a tool to assess whether the deformation exceeds the recoverable limit. The deformation gradient will be mapped back onto the boundary of elastic region using the following projection:

$$\mathcal{Z}(\mathbf{F}) = \begin{cases} \mathbf{F} & \delta\gamma \leq 0 \\ \mathbf{U}\exp(\boldsymbol{\epsilon} - \delta\gamma\frac{\hat{\boldsymbol{\epsilon}}}{\|\|\hat{\boldsymbol{\epsilon}}\|\|})\mathbf{V}^\mathsf{T} & \text{otherwise} \end{cases}, \tag{10}$$

where the $\delta\gamma = \|\hat{\boldsymbol{\epsilon}}\| - \frac{\tau_Y}{2\mu}$, $\boldsymbol{\epsilon} = \log(\Sigma)$ is the normalized Hencky strain. The $\mathbf{U}, \boldsymbol{\Sigma}$ and $\mathbf{V}$ can be obtained by performing Singular Value Decomposition (SVD) on deformation gradient $\mathbf{F}$.

**Granular Media**. Similar to plasticine, the StVK constitutive model is used to simulate granular media. Drucker-Prager yield criteria [48] is selected as the yielding condition. It is defined as follows:

$$\mathrm{Tr}(\boldsymbol{\epsilon}) > 0, \quad \text{or} \quad \delta\gamma = \|\hat{\boldsymbol{\epsilon}}\|_F + \alpha\frac{(d\lambda + 2\mu)\mathrm{Tr}(\boldsymbol{\epsilon})}{2\mu} > 0, \tag{11}$$

where $d$ is the spatial dimension, $\alpha = \sqrt{\frac{2}{3}}\frac{2\sin\theta_{fric}}{3-\sin\theta_{fric}}$ and $\theta_{fric}$ is the friction angle. The deformation gradient return mapping is defined by

$$\mathcal{Z}(\mathbf{F}) = \begin{cases} \mathbf{U}\mathbf{V}^\mathsf{T} & \mathrm{Tr}(\boldsymbol{\epsilon}) > 0 \\ \mathbf{F} & \delta\gamma \leq 0, \mathrm{Tr}(\boldsymbol{\epsilon}) \leq 0 \\ \mathbf{U}\exp\left(\boldsymbol{\epsilon} - \delta\gamma\frac{\hat{\boldsymbol{\epsilon}}}{\|\|\hat{\boldsymbol{\epsilon}}\|\|}\right)\mathbf{V}^\mathsf{T} & \text{otherwise} \end{cases}. \tag{12}$$

**Newtonian Fluid**. We adopt the approach used in PAC-NeRF [12], which employs a J-based fluid model combined with a viscosity term to simulate Newtonian fluids. The stress for this model is defined by

$$J\boldsymbol{\sigma} = \frac{1}{2}\mu(\nabla\mathbf{v} + \nabla\mathbf{v}^\mathsf{T}) + \kappa(J - \frac{1}{J^6}), \tag{13}$$

where $\mu$ and $\kappa$ represent the fluid viscosity and the bulk modulus, respectively.

**Non-Newtonian Fluid**. We employ the viscoplastic model [47] to simulate non-Newtonian fluids. Although we continue to utilize the von Mises criteria to delineate the elastic region, the presence of viscoplasticity implies that deformation will not be immediately reverted onto the yield surface. It is defined as follows:

$$
\mathcal{Z}(\mathbf{F}) = \begin{cases} \mathbf{F} & \delta\gamma \leq 0 \\ \mathbf{U}\exp(\frac{\hat{s}}{2\mu}\hat{\boldsymbol{\epsilon}} + \frac{1}{d}\mathrm{Tr}(\epsilon)\mathbf{I})\mathbf{V}^{\mathsf{T}} & \text{otherwise} \end{cases},
\tag{14}
$$

$$
\begin{aligned}
\hat{\mu} &= \frac{\mu}{d}\mathrm{Tr}(\boldsymbol{\Sigma}^2), \\
\boldsymbol{s} &= 2\mu\hat{\epsilon}, \\
\hat{s} &= \|\boldsymbol{s}\| - \frac{\delta\gamma}{1 + \frac{\eta}{2\hat{\mu}\Delta t}}
\end{aligned}
\tag{15}
$$

where $d$ is the spatial dimension. The $\mathbf{U}, \boldsymbol{\Sigma}$ and $\mathbf{V}$ can be obtained by performing Singular Value Decomposition (SVD) on deformation gradient $\mathbf{F}$.

