# OpenReview forum: "GIC: Gaussian-Informed Continuum for Physical Property Identification and Simulation"
_NeurIPS.cc/2024/Conference — NeurIPS 2024 oral_

### Official Review · Reviewer_mZdJ · 2024-07-10

**Soundness:** 3
**Presentation:** 2
**Contribution:** 3
**Rating:** 6
**Confidence:** 5

**Summary:**

The paper proposes an improvement of PAC-NeRF for the task of estimating material properties from multiview video using 3D Gaussian Splatting (3DGS). Instead of estimating geometry solely based on the first frame like PAC-NeRF, the proposed method uses 4D Gaussian Splatting (4DGS) with reduced order modeling to construct 4D geometry, enabling the use of 3D supervision. A coarse-to-fine internal filling strategy is introduced to ensure that the simulation operates on a solid volume. 2D mask loss is used for additional supervision.

**Strengths:**

The reduced order modeling of 4DGS is a good fit for the reconstruction task with a limited number of fixed views. Directly applying full-order 4DGS seems hard to optimize due to the high number of DOFs.

The experiment results are promising.

A real-world application is provided.

**Weaknesses:**

Some symbols are not defined clearly, making it hard to follow at times. For example, $Discretize$ operator in Line 214; $\tilde{P}$ and $F$ are not defined in the text. I need to guess from Alg 1.

**Questions:**

If the 4D geometry is accurate enough, is it adequate to only use 3D supervision? Ablation studies are needed to validate the necessity of 2D mask supervision.

Coarse-to-fine density field creation: At the beginning, the reconstruction contour is much larger than the object. How does it shrink to the actual boundary? The $TrilinearInterpolation$ operator will not shrink the contour, and there is no operator to assign zeros.

In the simulation, how do the Gaussian kernel scales evolve? The paper seems to assume isotropic kernels, but physical deformation can transform the sphere into an ellipse. How is this addressed?

**Limitations:**

Limitations are well discussed.

---

> ### Author Rebuttal · Authors · 2024-08-06
>
> We thank the reviewer for the detailed reading of our paper and constructive suggestions! We hope our responses adequately address the following questions about our work. Please let us know if there’s anything we can clarify further.
>
> ---
>
> > 1. Some symbols are not defined clearly, making it hard to follow at times. For example, $Discretize$ operator in Line 214; $\tilde{P}$ and $F$ are not defined in the text. I need to guess from Alg 1.
>
> **Reply:** Sorry for the lack of clarity. $Discretize$ denotes the operation mapping particle positions to voxel indices on the density field. \tilde{P}$ and $F$ are the sampled particles and the density field. We add a table to the attached PDF in the "Author Rebuttal" session to clarify the operators and symbols in detail. We will also add the table to the Appendix in the revised version.
>
> ---
>
> > 2. If the 4D geometry is accurate enough, is it adequate to only use 3D supervision? Ablation studies are needed to validate the necessity of 2D mask supervision.
>
> **Reply:** Thank you for this constructive advice.
>
> (1) To answer the first question, we perform system identification on the torus object, which is the only instance that provides a ground truth mesh model in the PAC-NeRF dataset. Specifically, we use the ground truth point clouds as the continuum for simulation and utilize the mesh model to extract surface particles from the point cloud as 3D supervision. The experiment, performed with this configuration and other settings unchanged, achieves $E = 1,002,907.25$ and $\mu = 0.2991$, which are close to the ground truth ($E = 1,000,000$ and $\mu = 0.3$). Therefore, we believe that **it should be sufficient to use 3D object surfaces as supervision once the recovered geometry is accurate enough**.
>
> (2) To evaluate the necessity of 2D mask supervision, we perform system identification on 45 cross-shaped object instances in the PAC-NeRF dataset by our method but with only object surface supervision. The results are reported in the table below. It is obvious to see that combining both 2D and 3D shapes as supervision can achieve more accurate performance compared to using 3D shapes only. Therefore, we believe that **utilizing 2D mask supervision to some extent makes up for the errors introduced by the 3D object surfaces extracted from dynamic 3D Gaussians**. We will add the analysis to the revised version.
>
> | **Type**          | **Parameters**   | **w/o masks**            | **w/ masks**             |
> |-------------------|------------------|--------------------------|--------------------------|
> | **Newtonian**     | $\log_{10}(\mu)$ | $2.19 \pm 2.90$          | $\mathbf{1.53 \pm 1.31}$ |
> |                   | $\log_{10}(\kappa)$ | $24.2 \pm 22.2$       | $\mathbf{14.8 \pm 19.2}$ |
> |                   | $v$              | $0.20 \pm 0.08$          | $\mathbf{0.20 \pm 0.07}$ |
> | **Non-Newtonian** | $\log_{10}(\mu)$ | $19.4 \pm 27.7$          | $\mathbf{13.5 \pm 18.2}$ |
> |                   | $\log_{10}(\kappa)$ | $24.0 \pm 24.8$       | $\mathbf{12.9 \pm 16.8}$ |
> |                   | $\log_{10}(\tau_Y)$ | $\mathbf{4.58 \pm 9.11}$ | $4.80 \pm 3.92$      |
> |                   | $\log_{10}(\eta)$ | $49.1 \pm 40.5$         | $\mathbf{40.7 \pm 24.6}$ |
> |                   | $v$              | $1.33 \pm 0.54$          | $\mathbf{0.19 \pm 0.09}$ |
> | **Elasticity**    | $\log_{10}(E)$   | $2.85 \pm 1.94$          | $\mathbf{2.43 \pm 3.29}$ |
> |                   | $\nu$            | $3.97 \pm 2.64$          | $\mathbf{2.52 \pm 2.03}$ |
> |                   | $v$              | $\mathbf{0.22 \pm 0.10}$ | $0.82 \pm 0.32$          |
> | **Plasticine**    | $\log_{10}(E)$   | $\mathbf{25.6 \pm 27.4}$ | $25.6 \pm 29.4$          |
> |                   | $\log_{10}(\tau_Y)$ | $9.04 \pm 2.37$       | $\mathbf{1.67 \pm 1.21}$ |
> |                   | $v$              | $1.16 \pm 0.00$          | $\mathbf{0.22 \pm 0.10}$ |
> | **Sand**          | $\theta_{fric}$  | $\mathbf{2.55 \pm 2.03}$ | $4.18 \pm 0.52$          |
> |                   | $v$              | $0.31 \pm 0.18$          | $\mathbf{0.17 \pm 0.05}$ |
>
> ---
>
> > 3. Coarse-to-fine density field creation: At the beginning, the reconstruction contour is much larger than the object. How does it shrink to the actual boundary? The $TrilinearInterpolation$ operator will not shrink the contour, and there is no operator to assign zeros.
>
> **Reply:** Sorry for the lack of clarity. Although all the operations cannot assign voxels outside an object to zeros, the trilinear interpolation and mean filter operations can reduce the density of the voxels outside the object boundary. By iteratively performing mean filtering and particle voxel reassigning, the densities outside the boundary will be sufficiently small while the object boundary and internal region keep high-density values, and we thus can extract the object particles by thresholding the density field.
>
> ---
>
> > 4. In the simulation, how do the Gaussian kernel scales evolve? The paper seems to assume isotropic kernels, but physical deformation can transform the sphere into an ellipse. How is this addressed?
>
> **Reply:** Sorry for the lack of clarity. In this work, we use the grid size of the density field as scale attributes of Gaussian kernels and fix them during the simulation. We admit that a physics-informed scale transformation such as PhysGaussian [1] allows a more realistic rendering. In future work, we will integrate this function into our method to enable kernel transformation during simulation.
>
> ---
>
> [1] Xie, Tianyi, et al. "Physgaussian: Physics-integrated 3d gaussians for generative dynamics." Proceedings of the IEEE/CVF Conference on Computer Vision and Pattern Recognition. 2024.

---

> > ### Comment · Reviewer_mZdJ · 2024-08-13
> >
> > Thank you for the rebuttal. I do not have further questions.

---

> > > ### Author Response · Authors · 2024-08-14
> > >
> > > Thanks for your feedback and we are pleased that our response has successfully addressed your concerns.

---

### Official Review · Reviewer_HGBq · 2024-07-12

**Soundness:** 2
**Presentation:** 2
**Contribution:** 2
**Rating:** 6
**Confidence:** 3

**Summary:**

This paper introduces a novel hybrid method that leverages 3D Gaussian representation and continuum to estimate physical properties of deformable objects. From multi-view video, the Gaussian- informed continuum can be extracted and then combined with material point method (MPM) simulation to train the whole pipeline by using both 3D shape and 2D as supervision. The experiments show that the proposed method outperforms previous approaches based on continuum dynamics or 3D Gaussian representation for dynamic reconstruction, system identification or other real-world applications.

**Strengths:**

1. The paper introduces an efficient motion-factorized dynamic 3D Gaussian network to reconstruct the object states as a linear combination of basis motions, in which the estimated motions and coefficients share the same backbone.

2. The generated Gaussian-informed continuum consists of the density and grid size scale given by the proposed coarse-to-fine filling strategy, which is further used as supervision in training together with the MPM simulation. This tackles the issue of using quantised Gaussian particles for simulation of continuous structures.

3. The experiments show that the method can achieve SoTA performance compared to pre- vious works among various deformable objects, especially when large deformation occurs. The method is moreover applicable to real-word scenarios.

**Weaknesses:**

1. The authors stated that such a lightweight architecture of the motion-factorized dynamic 3D Gaussian network is sufficient for complex motions rather than modeling each basis with an independent network (line 168-171) while lacking an experimental proof.

2. The representation in Section 4.3 is lacking in elaboration and should accompany with more details in the supplementary. Please give some detailed elaboration for Section 4.3 about Gaussian-informed continuum, e.g., notations used in Algorithm 1, dimensions of the variables, etc.

**Questions:**

1. It would be nice to see a  comparison between the choice of Motion-factorized dynamic 3D Gaussian network and previous architectures ?

2. It is important that the experimentation on complex motions must be performed in order to truly understand the strength of the proposed  method.

**Limitations:**

To some extent, yes.

---

> ### Author Rebuttal · Authors · 2024-08-06
>
> We thank the reviewer for the detailed reading of our paper and constructive suggestions! We hope our responses adequately address the following questions about our work. Please let us know if there’s anything we can clarify further.
>
> ---
>
> > 1. The authors stated that such a lightweight architecture of the motion-factorized dynamic 3D Gaussian network is sufficient for complex motions rather than modeling each basis with an independent network (line 168-171) while lacking an experimental proof.
>
> **Reply:** Thank you for pointing out this question. We conducted two ablation analyses to empirically demonstrate the arguments mentioned in lines 168-171.
>
> (1) We first use our method to perform dynamic Gaussian reconstructions on 45 cross-shaped objects in the PAC-NeRF dataset, except each motion basis is modeled with an independent network. Specifically, each basis takes the encoded time as input and contains 8 fully connected layers. The output is the residuals of position and scale of this basis. We use the setting mentioned in Sec. 5.1 to evaluate the CD and EMD on the above method variants and compare them with our method. The results are reported in the table below. The results show that our backbone-shared architecture (**Ours**) slightly outperforms the independent-basis networks (**Ind.**) in terms of dynamic reconstruction, which empirically demonstrates the strength that the reduced order modeling of dynamic Gaussians is sufficient for motion reconstruction tasks. Moreover, our method can achieve less training time compared with the independent design (around 15 minutes vs. 45 minutes for a dynamic scene on a single 3090 GPU).
>
> |                | CD $\downarrow$ (Ind.)    | CD $\downarrow$ (**Ours**) | EMD $\downarrow$ (Ind.)    | EMD $\downarrow$ (**Ours**) |
> |----------------|---------------------------|----------------------------|----------------------------|-----------------------------|
> | Newtonian      | 0.250                     | **0.243**                  | 0.026                      | **0.025**                   |
> | Non-Newtonian  | 0.204                     | **0.195**                  | 0.023                      | **0.022**                   |
> | Elasticity     | 0.188                     | **0.178**                  | 0.022                      | **0.020**                   |
> | Plasticine     | 0.215                     | **0.196**                  | 0.024                      | **0.022**                   |
> | Sand           | 0.273                     | **0.250**                  | 0.028                      | **0.025**                   |
> | Mean           | 0.226                     | **0.212**                  | 0.025                      | **0.023**                   |
>
> (2) To validate the second argument in lines 171-172, we compared our method with DynMF [1], which also uses neural networks as learnable bases while considering motion coefficients as time-invariant Gaussian attributes, by evaluating the PSNR on the D-NeRF dataset. The results are reported in the table below. The results show that modeling the motion coefficients as time-variant variables does increase the ability to fit the dynamic scenes.
>
> | Method    | Hell Warrior | Mutant | Hook  | Bouncing Balls | T-Rex  | Stand Up | Jumping Jacks | Mean  |
> |-----------|--------------|--------|-------|----------------|--------|----------|---------------|-------|
> | DynMF [1] | 36.60        | 41.00  | 31.30 | 41.01          | 35.10  | 41.16    | 35.75         | 37.42 |
> | Ours      | **41.97**    | **42.93** | **38.04** | **41.26** | **37.54** | **45.32** | **38.86** | **40.85** |
>
> ---
>
> > 2. The representation in Section 4.3 is lacking in elaboration and should accompany with more details in the supplementary. Please give some detailed elaboration for Section 4.3 about Gaussian-informed continuum, e.g., notations used in Algorithm 1, dimensions of the variables, etc.
>
> **Reply:** Thank you for this constructive advice. As suggested, we added a table to the attached PDF in the "Author Rebuttal" session to clarify the operators and symbols in detail. We will also add the table to the Appendix in the revised version.
>
> ---
>
> > 3. It would be nice to see a comparison between the choice of Motion-factorized dynamic 3D Gaussian network and previous architectures ?
>
> **Reply:** Please refer to the first reply.
>
> ---
>
> > 4. It is important that the experimentation on complex motions must be performed in order to truly understand the strength of the proposed method.
>
> **Reply:** Thank you for this suggestion. We conduct an additional experiment on a scenario with a more complex boundary condition and motion trajectory. Specifically, we use our method to perform system identification on an elastic rope falling onto two rigid cylinders. The data format is the same as PAC-NeRF. The estimated property is reported in the table below, and the simulated trajectory is visualized in the attached PDF. The results show that our method can also generalize to scenarios with more complex boundary conditions and motions.
>
> |        | Initial Guess | PacNeRF         | Ours               | Ground Truth |
> |--------|---------------|-----------------|--------------------|--------------|
> | $E$    | $10^3$        | $1.12 \times 10^5$ | $\mathbf{1.03 \times 10^5}$ | $10^5$       |
> | $\nu$  | $0.4$         | $0.22$          | $\mathbf{0.23}$    | $0.3$        |
>
> ---
>
> [1] Kratimenos, Agelos, Jiahui Lei, and Kostas Daniilidis. "Dynmf: Neural motion factorization for real-time dynamic view synthesis with 3d gaussian splatting." arXiv preprint arXiv:2312.00112 (2023).

---

> > ### Comment · Reviewer_HGBq · 2024-08-12
> >
> > I thank the authors for the rebuttal.
> >
> > I don't have any questions at the moment.

---

> > > ### Author Response · Authors · 2024-08-13
> > >
> > > Thanks for the reviewer's feedback. We are pleased that our response has addressed the concerns. We would appreciate that if the reviewer could re-evaluate the review score. (Note: since the openreview system had issue earlier that reviewers cannot receive the email after posting comments, we delete the old comment and resend it.)

---

### Official Review · Reviewer_a7go · 2024-07-13

**Soundness:** 3
**Presentation:** 3
**Contribution:** 3
**Rating:** 7
**Confidence:** 3

**Summary:**

The manuscript proposes a novel hybrid framework that leverages 3D Gaussian representations for system identification from visual observations. The framework captures both explicit and implicit shapes using dynamic 3D Gaussian reconstruction and a coarse-to-fine filling strategy to generate density fields. These fields are used to sample continuum particles for simulation and extract object surfaces, which can render object masks during simulations to guide physical property estimation.

**Strengths:**

1. The presentation is clear. The figures look high-quality.
2. I personally appreciate the real-world experiments. I am happy to see the proposed method also works in real life.
3. The proposed method solves one of the most interesting problem in the intersection of guassian splatting and physical simulation, where physical simulation requires volumetric representation but 3dgs outputs surfaces.

**Weaknesses:**

1. In the real-world experiment, I found the authors switched to FEM for deformable body simulation, which conflicts with the MPM simulator used in their pipeline. I think it needs justifications.
2. Some wordings are confusing: e.g., line 166, do you mean effective instead of efficient?

**Questions:**

1. I wonder what makes the difference between the proposed method and PAC-NeRF in the infilled particle generation.
2. If I understood correctly, the motion network only takes time as input. I wonder if it helps by combining the temporal encoding from DiffAqua [1] to further capture the low and high frequencies.
3. The infilling algorithm seems expensive since the complexity grows exponentially. Did you try using Octree [2] or similar algorithm to speed it up?

[1] Ma, Pingchuan, et al. "Diffaqua: A differentiable computational design pipeline for soft underwater swimmers with shape interpolation." ACM Transactions on Graphics (TOG) 40.4 (2021): 1-14.

[2] Meagher, Donald JR. Octree encoding: A new technique for the representation, manipulation and display of arbitrary 3-d objects by computer. Electrical and Systems Engineering Department Rensseiaer Polytechnic Institute Image Processing Laboratory, 1980.

**Limitations:**

Yes.

---

> ### Author Rebuttal · Authors · 2024-08-06
>
> We thank the reviewer for the detailed reading of our paper and constructive suggestions! We hope our responses adequately address the following questions about our work. Please let us know if there’s anything we can clarify further.
>
> ---
>
> > 1. In the real-world experiment, I found the authors switched to FEM for deformable body simulation, which conflicts with the MPM simulator used in their pipeline. I think it needs justifications.
>
> **Reply:** Sorry for the confusion caused. Switching to FEM is because most of the widely-used robotic simulators incorporate FEM for deformable object simulation, including Isaac Gym, the simulator we used in our experiments. Moreover, estimating physical parameters using MPM and then applying them to FEM should be a feasible way for this application, since
>
> (1) both the Material Point Method (MPM) and the Finite Element Method (FEM) originate from Galerkin methods [3]. In theory, MLS-MPM can achieve first-order consistency in simulations [3], which is comparable to the consistency achieved by linear FEM simulations,
>
> (2) the material properties (e.g., Young's modulus and Poisson's ratio) are independent of the numerical methods.
>
> We will clarify this in the revised version.
>
> ---
>
> > 2. Some wordings are confusing: e.g., line 166, do you mean effective instead of efficient?
>
> **Reply:** Sorry for the confusion caused. Indeed, we should use "effective" in line 166. We will correct the expression in the revised version.
>
> ---
>
> > 3. I wonder what makes the difference between the proposed method and PAC-NeRF in the infilled particle generation.
>
> **Reply:** Sorry for the lack of clarity.
>
> (1) PAC-NeRF samples particles by directly selecting NeRF field voxels whose alpha values are greater than the predefined threshold and then performing uniform sampling 4 times on each extracted voxel. Since PAC-NeRF defined the alpha threshold with **a small value** to make sure that a solid continuum can be obtained, they usually tend to recover over-large shapes.
>
> (2) On the contrary, our method turns to generate a density field based on the Gaussian and initial particles {$\mu(t)$} $\cup P_{in}$. Among these particles, the Gaussian particles are prone to be located at the object surface, which can guarantee high density at the surface region, while the initial ones incorporated with the coarse-to-fine operations can ensure a solid internal region. With this module, we can extract the continuum from the density field, which serves only for **representing the object shape** instead of the NeRF field, which is utilized for **rendering**. Please refer to Figure 7 in Appendix A.2.2 to see the qualitative results of the proposed filling algorithm, along with ones from PAC-NeRF.
>
> We will add more explanation in the revised version.
>
> ---
>
> > 4. If I understood correctly, the motion network only takes time as input. I wonder if it helps by combining the temporal encoding from DiffAqua [1] to further capture the low and high frequencies.
>
> **Reply:** Sorry for the lack of clarity. We do employ temporal and positional encoding to the time $t$ and position $\mu_0$, respectively, to introduce features with various frequencies. Specifically, the encoding module is denoted as $\gamma(x) = \left( \sin(2^k \pi x), \cos(2^k \pi x) \right)_{k=0}^{L-1}$, where $L=10$ for both $t$ and $\mu_0$, which is exactly the same as the setting in DiffAqua. We will add the notations to Figure 2 and the implementation details to Appendix A.1.1 in the revised version.
>
> ---
>
> > 5. The infilling algorithm seems expensive since the complexity grows exponentially. Did you try using Octree [2] or similar algorithm to speed it up?
>
> **Reply:** Although the memory requirements for processing the volumetric data scales cubically with the grid resolution, we still use the naive volumetric data structure for our algorithm because
>
> (1) in practice, only **four** iterations are required to achieve sufficient accuracy (the implementation details are also available in Appendix A.2.1),
>
> (2) such a data structure can be efficiently implemented with GPU acceleration based on PyTorch, where the trilinear interpolation and mean filter operations are implemented by "grid_sample" and "conv3d" functions, respectively.
>
> Therefore, we can almost achieve real-time performance (more than 10 fps on a single Nvidia 3090 GPU) on our infilling algorithm.
>
> ---
>
> [3] Hu, Yuanming, et al. "A moving least squares material point method with displacement discontinuity and two-way rigid body coupling." ACM Transactions on Graphics (TOG) 37.4 (2018): 1-14.

---

> > ### Comment · Reviewer_a7go · 2024-08-10
> >
> > I appreciate the authors' responses, which addressed most of my concerns and questions. Regarding the first question, I suggest moderating the stance on the orthogonality between the numerical method and the corresponding physical parameters. While the claim is theoretically sound, practitioners often encounter significant misalignment between MPM and FEM, and I fear this statement might be misleading. It would be beneficial to see dedicated work on FEM due to its realism and applicability in robotics-related tasks. However, I understand this would require considerable separate effort and merits its own publication. Therefore, I will raise my score to 7 to advocate for acceptance.

---

> > > ### Author Response · Authors · 2024-08-12
> > >
> > > Thanks for the reviewer's feedback. We are pleased that our response has addressed the concerns. We admit that although the orthogonality assumption works on our application, there might be practical challenges when aligning MPM and FEM, especially on more complex tasks. We'll moderate our stance in the revised version.

---

### Official Review · Reviewer_FGDU · 2024-07-17

**Soundness:** 3
**Presentation:** 3
**Contribution:** 3
**Rating:** 7
**Confidence:** 4

**Summary:**

This paper presents an approach for estimating the geometry and physical properties of objects through visual observations using 3D Gaussian representations. The method employs a dynamic 3D Gaussian framework to reconstruct objects as point sets over time and a coarse-to-fine filling strategy to generate density fields. This facilitates the extraction of object continuums and integrates Gaussian attributes, aiding in rendering object masks during simulations for implicit shape guidance.The extracted geometries are then used to guide physical property estimation through differentiable MPM simulation.

**Strengths:**

The experiments in this paper demonstrate improvements over prior works such as PAC-NeRF and Spring-Gaus. The introduction of a novel hybrid framework leveraging 3D Gaussian representations for physical property estimation is straightforward and easy to understand, and experiments have confirmed their effectiveness.  Overall, the paper makes meaningful contributions to the problem of geometry + physical property estimation from multi-view videos.

**Weaknesses:**

Some of the technical terms are misused, making the paper confusing to read. For example, "implicit shape representation" is repeatedly used to refer to the rendered object image masks, but "implicit" generally refers to using functions (parametric or neural networks) to represent shapes, where shapes must be retrieved through function evaluations, hence "implicit". Please correct this terminology, as the geometries in this work are represented using GS, which are explicit representations. Additionally, some of the results presentations are confusing and could benefit from clearer explanations and more organized presentation (see below). Clarifying these aspects would greatly enhance the paper's readability and overall impact.

**Questions:**

1. Table 3: Can you include the ground truth values in the table so that readers understand what to expect?
2. System Identification: How do you set the initial parameters for the system identification? Are the optimizations sensitive to initial conditions?
3. Figure 4: Would mask-based supervision fail when the estimated shapes are significantly different from the ground truth? Have you observed any cases where this occurs?
4. Figure 1a: There is a typo in the figure caption. Change "caption" to "capture."
5. References: It would be beneficial to include references to works on differentiable cloth simulation for system identification and inverse problems, such as "Differentiable Cloth Simulation for Inverse Problems" by Liang et al. and "DiffCloth: Differentiable Cloth Simulation with Dry Frictional Contact" by Li et al.

**Limitations:**

The authors have discussed the limitations of the method regarding known camera parameters, assumption of known material models, and continuum mechanics. However, I do wonder about the failure cases of the method, if there are any. Understanding specific scenarios where the method does not perform well would provide valuable insights and help guide future improvements.

---

> ### Author Rebuttal · Authors · 2024-08-06
>
> We thank the reviewer for the detailed reading of our paper and constructive suggestions! We hope our responses adequately address the following questions raised about our work. Please let us know if there’s anything we can clarify further.
>
> ---
>
> > 0. Some of the technical terms are misused, making the paper confusing to read. For example, "implicit shape representation" is repeatedly used to refer to the rendered object image masks, but "implicit" generally refers to using functions (parametric or neural networks) to represent shapes, where shapes must be retrieved through function evaluations, hence "implicit". …
>
> **Reply:** Sorry for the confusion caused. Since our method employs both 3D surface particles and 2D object masks for supervision, we originally intended to use "implicit shape representation" to describe 2D object masks in order to distinguish them from 3D surface particles, but we overlooked the ambiguity it introduced. We will correct the description by directly using 2D object masks in the revised version.
>
> ---
>
> > 1. Table 3: Can you include the ground truth values in the table so that readers understand what to expect?
>
> **Reply:** Thanks for this constructive advice. For more details about the estimated and ground truth values, please refer to the attached PDF in the "Author Rebuttal" session. We will also add the table to the Appendix in the revised version.
>
> ---
>
> > 2. System Identification: How do you set the initial parameters for the system identification? Are the optimizations sensitive to initial conditions?
>
> **Reply:** The table in the attached PDF also lists each instance's initial guess. To make a fair comparison, we followed the setting in PAC-NeRF [1] to assign the same initial values for the instances with the same material. The results show that our method is robust to initial conditions even when they are significantly different from the ground truth.
>
> ---
>
> > 3. Figure 4: Would mask-based supervision fail when the estimated shapes are significantly different from the ground truth? Have you observed any cases where this occurs?
>
> **Reply:** Under extreme conditions, the mask-based supervision might fail when the simulated trajectory is completely out of view for all viewpoints. However, we did not find any failure cases in our experiments, because we performed initial velocity estimation before system identification. With the initial velocity available and multiple viewpoints located at proper positions, we observed that it's unlikely that the simulated trajectory is outside the field of view, and the estimated shapes will always converge to the ground truth shapes even if they have significant discrepancies at the initial stage.
>
> ---
>
> > 4. Figure 1a: There is a typo in the figure caption. Change "caption" to "capture.”
>
> **Reply:** Thank you for pointing this out. We will correct the typo in the revised version.
>
> ---
>
> > 5. References: It would be beneficial to include references to works on differentiable cloth simulation for system identification and inverse problems, such as "Differentiable Cloth Simulation for Inverse Problems" by Liang et al. and "DiffCloth: Differentiable Cloth Simulation with Dry Frictional Contact" by Li et al.
>
> **Reply:** Thank you for this suggestion. These two works also tackle the inverse problems with differentiable simulators, particularly in cloth material. We will cite the related works you mentioned in our revised manuscript.
>
> ---
>
> > 6. Limitations: I do wonder about the failure cases of the method if there are any.
>
> **Reply:** In the synthetic experiments, we didn't encounter any failure cases on both the PAC-NeRF [1] and SpringGaus [2] synthetic datasets, since each scenario includes video sequences from 10 or 11 distinct viewpoints, which are sufficient for dynamic Gaussian reconstruction. However, the proposed dynamic Gaussian module would fail to reconstruct the trajectory on the SpringGaus real dataset because it only contains 3 viewpoints to capture the dynamic scene. That's why we only use object masks for the supervision of system identification (More details are illustrated in Section 5.3 and Appendix A.5.). Therefore, our method might not perform well when **fewer views** of the scenario are available. Using fewer views to recover both geometry and system identification is more practical and will be an interesting direction for future work.
>
> ---
>
> [1] Xuan Li, Yi-Ling Qiao, Peter Yichen Chen, Krishna Murthy Jatavallabhula, Ming Lin, Chen-fanfu Jiang, and Chuang Gan. Pac-nerf: Physics augmented continuum neural radiance fields for geometry-agnostic system identification. In Proceedings of the International Conference on Learning Representations (ICLR), 2022.
>
> [2] Licheng Zhong, Hong-Xing Yu, Jiajun Wu, and Yunzhu Li. Reconstruction and simulation of elastic objects with spring-mass 3d gaussians. arXiv preprint arXiv:2403.09434, 2024.

---

> > ### Comment · Reviewer_FGDU · 2024-08-13
> >
> > Thank you for the reply. The replies addressed my concerns and there I'll raise my score.

---

> > > ### Author Response · Authors · 2024-08-14
> > >
> > > Thanks for your feedback and we are pleased that our response has successfully addressed your concerns.

---

### Author Rebuttal · Authors · 2024-08-06

We would like to express our sincere gratitude to all the reviewers for their time and their valuable feedback. We deeply appreciate their recognition of our work, such as
"The experiments in this paper demonstrate improvements over prior works" (FGDU),
"I am happy to see the proposed method also works in real life" (a7go),
"The proposed method solves one of the most interesting problems in the intersection of Gaussian splatting and physical simulation" (a7go),
"The experiments show that the method can achieve SoTA performance" (HGBq), and
"The reduced order modeling of 4DGS is a good fit for the reconstruction task" (mZdJ).
We hope that our work indeed "makes meaningful contributions to the problem of geometry + physical property estimation" (FGDU).

Inspired by their thoughtful comments, we have incorporated the following changes in the revision of our paper:

- We conducted four additional experiments, including
	- a comparison of our proposed network and independent basis baselines in terms of dynamic reconstruction,
	- system identification on a rope instance with more complex motion and boundary conditions,
	- system identification on the torus instance with ground truth point cloud and surface supervision, and
	- system identification on 45 cross-shaped object instances with only 3D surface supervision,
	to address the concerns of the reviewers.
- We updated our manuscript to fix typos and misused terminology to reduce the potential for misunderstandings.
- We added a table to the appendix to provide the estimated and ground truth values for Table 3 in the main manuscript (see Figure 1 in the attached PDF).
- We added a table to the appendix to clarify the operators and symbols in Algorithm 1 in detail (see Figure 2 in the attached PDF).

We hope our responses adequately address the questions raised about our work. Please let us know if there is anything else we can clarify further.

---

### Decision · Program_Chairs · 2024-09-25

**Decision:**

Accept (oral)

**Comment:**

This paper has received unanimous acceptance recommendation (2 WA, 2A), it is addressing an important problem and shows potential for real world applications. It is recommended for oral presentation for its high impact.